# Mechanism of ubiquitin ligation and lysine prioritization by a HECT E3

Hari B Kamadurai[1]*, Yu Qiu[1†], Alan Deng[1†‡], Joseph S Harrison[2,3], Chris MacDonald[4], Marcelo Actis[5], Patrick Rodrigues[6], Darcie J Miller[1], Judith Souphron[7], Steven M Lewis[2], Igor Kurinov[8], Naoaki Fujii[5], Michal Hammel[9], Robert Piper[4], Brian Kuhlman[2], Brenda A Schulman[1,10]*

[1]Department of Structural Biology, St Jude Children's Research Hospital, Memphis, United States; [2]Department of Biochemistry and Biophysics, University of North Carolina at Chapel Hill, Chapel Hill, United States; [3]Lineberger Comprehensive Cancer Center, University of North Carolina at Chapel Hill, Chapel Hill, United States; [4]Department of Molecular Physiology and Biophysics, University of Iowa, Iowa City, United States; [5]Department of Chemical Biology and Therapeutics, St Jude Children's Research Hospital, Memphis, United States; [6]Hartwell Center for Bioinformatics and Biotechnology, St Jude Children's Research Hospital, Memphis, United States; [7]Institut Curie, CNRS UMR 3306, INSERM U1005, Orsay, France; [8]Department of Chemistry and Chemical Biology, Cornell University, Argonne, United States; [9]Physical Biosciences Division, Lawrence Berkeley National Laboratory, Berkeley, United States; [10]Howard Hughes Medical Institute, St Jude Children's Research Hospital, Memphis, United States

*For correspondence: hari.kamadurai@stjude.org (HBK); brenda.schulman@stjude.org (BAS)

†These authors contributed equally to this work

‡Present address: Department of Chemistry, Stanford University, Palo Alto, United States

Competing interests: The authors declare that no competing interests exist.

**Abstract** Ubiquitination by HECT E3 enzymes regulates myriad processes, including tumor suppression, transcription, protein trafficking, and degradation. HECT E3s use a two-step mechanism to ligate ubiquitin to target proteins. The first step is guided by interactions between the catalytic HECT domain and the E2~ubiquitin intermediate, which promote formation of a transient, thioester-bonded HECT~ubiquitin intermediate. Here we report that the second step of ligation is mediated by a distinct catalytic architecture established by both the HECT E3 and its covalently linked ubiquitin. The structure of a chemically trapped proxy for an E3~ubiquitin-substrate intermediate reveals three-way interactions between ubiquitin and the bilobal HECT domain orienting the E3~ubiquitin thioester bond for ligation, and restricting the location of the substrate-binding domain to prioritize target lysines for ubiquitination. The data allow visualization of an E2-to-E3-to-substrate ubiquitin transfer cascade, and show how HECT-specific ubiquitin interactions driving multiple reactions are repurposed by a major E3 conformational change to promote ligation.

## Introduction

A prevailing mechanism for altering protein function involves post-translational modification by ubiquitin (Ub) via E1-E2-E3 trienzyme cascades. First, an E1 activating enzyme catalyzes formation of a transient E2~Ub intermediate, which is linked by a thioester bond between Ub's C-terminus and the catalytic cysteine of an E2 conjugating enzyme (here, '~' refers to a covalent protein–protein interaction linked by a thioester, oxyester, or isopeptide bond, '-' refers to a noncovalent interaction, and 'x' refers to a crosslinked complex). E3s subsequently utilize one of two general mechanisms to adjoin the activated Ub C-terminus with targets. E3s in the RING family promote transfer of Ub's C-terminus from the E2 catalytic cysteine to a substrate (*Deshaies and Joazeiro, 2009*). Notably,

**eLife digest** Ubiquitin is a small protein that can be covalently linked to other, 'target', proteins in a cell to influence their behavior. Ubiquitin can be linked to its targets either as single copies or as polyubiquitin chains in which several ubiquitin molecules are bound end-on-end to each other, with one end of the chain attached to the target protein. A multi-step cascade involving enzymes known as E1, E2, and E3 adds ubiquitin to its targets. These enzymes function in a manner like runners in a relay, with ubiquitin a baton that is passed from E1 to E2 to E3 to the target.

The E3 enzyme is a ligase that catalyzes the formation of a new chemical bond between a ubiquitin and its target. There are approximately 600 different E3 enzymes in human cells that regulate a wide variety of target proteins. A major class of E3 enzymes, called HECT E3s, attaches ubiquitin to its targets in a unique two-step mechanism: the E2 enzymes covalently link a ubiquitin to a HECT E3 to form a complex that subsequently transfers the ubiquitin to its target protein. The ubiquitin is typically added to a particular amino acid, lysine, on the target protein, but the details of how HECT E3s execute this transfer are not well understood. To address this issue, Kamadurai et al. investigate how Rsp5, a HECT E3 ligase in yeast, attaches ubiquitin to a target protein called Sna3.

All HECT E3s have a domain—the HECT domain—that catalyzes the transfer of ubiquitin to its target protein. This domain consists of two sub-structures: the C-lobe, which can receive ubiquitin from E2 and then itself become linked to ubiquitin, and the N-lobe. These lobes were previously thought to adopt various orientations relative to each other to deliver ubiquitin to sites on different target proteins (including to multiple lysines on a single target protein). Unexpectedly, Kamadurai et al. find that in order to transfer the ubiquitin to Sna3, Rsp5 adopts a discrete HECT domain architecture that creates an active site in which parts of the C-lobe and the N-lobe, which are normally separated, are brought together with a ubiquitin molecule. This architecture also provides a mechanism that dictates which substrate lysines can be ubiquitinated based on how accessible they are to this active site.

The same regions of Rsp5 transfer ubiquitin to targets other than Sna3, suggesting that a uniform mechanism—which Kamadurai et al. show is conserved in two related human HECT E3 ligases—might transfer ubiquitin to all its targets. These studies therefore represent a significant step toward understanding how a major class of E3 enzymes modulates the functions of their targets.

recent structural studies have revealed how isolated RING domains bind both E2 and Ub to optimally orient and activate the E2~Ub intermediate for nucleophilic attack (*Dou et al., 2012*; *Plechanovová et al., 2012*; *Pruneda et al., 2012*). Many ligases utilize a different mechanism possessing an active site cysteine participating directly in catalysis via a two-step mechanism. First, Ub is transferred from an E2~Ub intermediate to the E3 catalytic cysteine to form a labile, thioester-linked E3~Ub intermediate. Second, Ub is transferred from the E3 cysteine to a substrate's primary amino group, which is typically from a lysine side-chain. This catalytic mechanism is common to many classes of E3s, including several effector proteins from bacterial pathogens, the RING-IBR-RING family, and HECT (homologous to E6AP C-terminus) ligases (*Scheffner et al., 1995*; *Zhang et al., 2006*; *Rohde et al., 2007*; *Wenzel et al., 2011*). Despite the importance of thioester-forming E3s, the mechanisms by which Ub is ligated from an E3 cysteine remain elusive, because at the time of original manuscript submission there were no structures mimicking an E3~Ub intermediate, and to date there are none representing an E3~Ub-substrate complex.

HECT ligases were among the first family of E3 enzymes to be cloned, and are the best functionally characterized among the thioester-forming E3s (*Huibregtse et al., 1995*; *Scheffner et al., 1995*). Deregulation of HECT E3-mediated ubiquitination is associated with diseases such as cancers, neurological disorders, autoimmunity, and hypertension (reviewed in *Bernassola et al., 2008*; *Rotin and Kumar, 2009*; *Metzger et al., 2012*; *Scheffner and Kumar, 2013*). Thus, it is important to understand the mechanisms underlying ubiquitination by E3s in the HECT family.

The common feature of HECT E3s is the ~40 kDa C-terminal catalytic HECT domain. Isolated C-terminal HECT catalytic domains are capable of binding E2~Ub, forming an E3~Ub intermediate, and ligating the E3-linked 'donor' Ub to a specific lysine on an 'acceptor' Ub to build a polyubiquitin chain (reviewed in *Kee and Huibregtse, 2007*). Accessory domains upstream of the HECT domain

recruit specific substrates for Ub ligation. The N-terminal sequences guide the classification of HECT E3s into different sub-families, the largest of which is the NEDD4-family. NEDD4-family members share a common domain organization with an N-terminal membrane binding C2 domain, two to four substrate-binding WW domains, and a C-terminal HECT domain. This modular architecture enables regulation and interactions (reviewed in *Ingham et al., 2004*; *Kee and Huibregtse, 2007*; *Rotin and Kumar, 2009*). Substrates often contain recognition sequences that bind directly to one or more WW domains. For example, PPXY (Pro-Pro-X-Tyr) is one common targeting sequence. Some HECT substrates are multiprotein complexes, with the PPXY motif and ubiquitination sites located in different polypeptides. Also, many substrates of NEDD4-family E3s are transmembrane proteins in which the PPXY motif and ubiquitination sites reside in separate cytosolic segments. Furthermore, some substrates do not bind directly to NEDD4-family E3s, but are instead recruited by adaptor proteins bearing PPXY-like sequences (reviewed in *Léon and Haguenauer-Tsapis, 2009*). In addition, for NEDD4-family E3s, a noncovalent ubiquitin binding site was discovered in the HECT domain, which promotes polyubiquitination through poorly understood mechanisms (*French et al., 2009*; *Ogunjimi et al., 2010*; *Kim et al., 2011*; *Maspero et al., 2011*).

The catalytic HECT domain consists of an N-lobe that binds E2, a flexible linker, and a C-lobe containing the active site cysteine (*Huang et al., 1999*; *Verdecia et al., 2003*). A prior structure of an E2~Ub-HECT intermediate revealed that Ub transfer from an E2 to a NEDD4-family HECT E3 is directed by noncovalent interactions between the HECT domain N-lobe and E2, and by contacts between the HECT domain C-lobe and the E2-linked Ub (*Kamadurai et al., 2009*). A structure published while the present work was under consideration showed preservation of this HECT domain-Ub conformation immediately upon generation of the E3~Ub intermediate (*Kamadurai et al., 2009*; *Maspero et al., 2013*), but raised the question of how the E3~Ub thioester bond could be directed to a substrate. Although structural mechanisms underlying Ub ligation from the E3 to a substrate remain poorly understood, a prevailing model is that numerous different relative orientations of the N- and C-lobes, manifested in prior crystal structures of HECT domains, enable Ub ligation at spatially distinct sites associated with polyubiquitination of diverse substrates (*Huang et al., 1999*; *Verdecia et al., 2003*; *Ogunjimi et al., 2005*). However, to date, this hypothesis has not been experimentally tested, and how Ub and substrates are positioned into the catalytic domain and how the target lysines are prioritized after their initial capture remains unknown.

To address the mechanisms of HECT E3-mediated Ub ligation, we studied Rsp5, the single essential NEDD4-family member from *Saccharomyces cerevisiae*. Rsp5 is required for proteasomal processing of a membrane-bound transcription factor (*Hoppe et al., 2000*). Rsp5 also functions in a multitude of diverse ubiquitin-dependent pathways including 26S proteasomal degradation of RNA polymerase II, and mediates proteasome-independent regulation of membrane protein trafficking (reviewed in *Ingham et al., 2004*; *Kaliszewski and Zoladek, 2008*; *Rotin and Kumar, 2009*). Rsp5 has a typical NEDD4-family domain structure consisting of a C2 domain, three WW domains, and a HECT domain. Although the Rsp5 C2 domain is required for regulation of endocytosis, a construct comprising only the third WW and HECT domains (hereafter referred to as 'Rsp5$^{WW3-HECT}$') is sufficient to support viability under normal growth conditions (*Springael et al., 1999*; *Hoppe et al., 2000*). Rsp5 substrates are typically modified either by a single Ub ('monoUb') or by polyubiquitin chains containing isopeptide linkages between Lys63 and the C-terminus of the subsequent Ub in the chain (*Shih et al., 2003*; *Kim and Huibregtse, 2009*; *Wilson et al., 2012*). Several prior studies have suggested that the Rsp5 substrate Sna3, a small transmembrane protein that undergoes Ub-dependent sorting into the multivesicular body pathway, would be a good model for studying HECT E3-mediated Ub ligation, because both the Rsp5-binding sequence and a major ubiquitination site are encompassed in Sna3's soluble C-terminal cytoplasmic domain (*McNatt et al., 2007*; *Oestreich et al., 2007*; *Stawiecka-Mirota et al., 2007*; *Watson and Bonifacino, 2007*; *Macdonald et al., 2012*). Rsp5 binds via its third WW domain to a PPXY motif (residues 106–109) in Sna3, and ubiquitinates Sna3 at Lys125 (*McNatt et al., 2007*; *Oestreich et al., 2007*; *Stawiecka-Mirota et al., 2007*; *Watson and Bonifacino, 2007*; *Macdonald et al., 2012*).

Here we define mechanisms underlying HECT E3-mediated Ub ligation. We report the crystal structure of a ternary complex in which Rsp5's active site is simultaneously crosslinked to both Ub's C-terminus and Sna3, as well as biochemical, genetic, and molecular modeling experiments. Together, the data reveal interactions between the HECT domain N-lobe, the HECT domain C-lobe, and its thioester-linked Ub, establishing a specific architecture for ligation. This conformation both confines

the selection of target lysines and displays a composite active site wherein both HECT domain lobes contribute to catalysis of ubiquitin transfer.

## Results

### A simplified HECT E3 ligation model system

To facilitate biophysical and structural studies, we identified a minimal HECT E3-substrate pair (*Figure 1*). For an Rsp5 substrate, we turned to the cytoplasmic domain of Sna3 (hereafter Sna3$^C$), because peptide-like substrates have proven to be very useful for providing insights into the ubiquitin ligation mechanisms of other families of E3 ligases. The corresponding minimal region of Rsp5 mediating Sna3$^C$ ubiquitination was identified through deletion mapping using a pulse-chase assay. Briefly, a thioester-bonded E2~Ub intermediate was enzymatically generated using E1, the E2 UbcH5B, and a

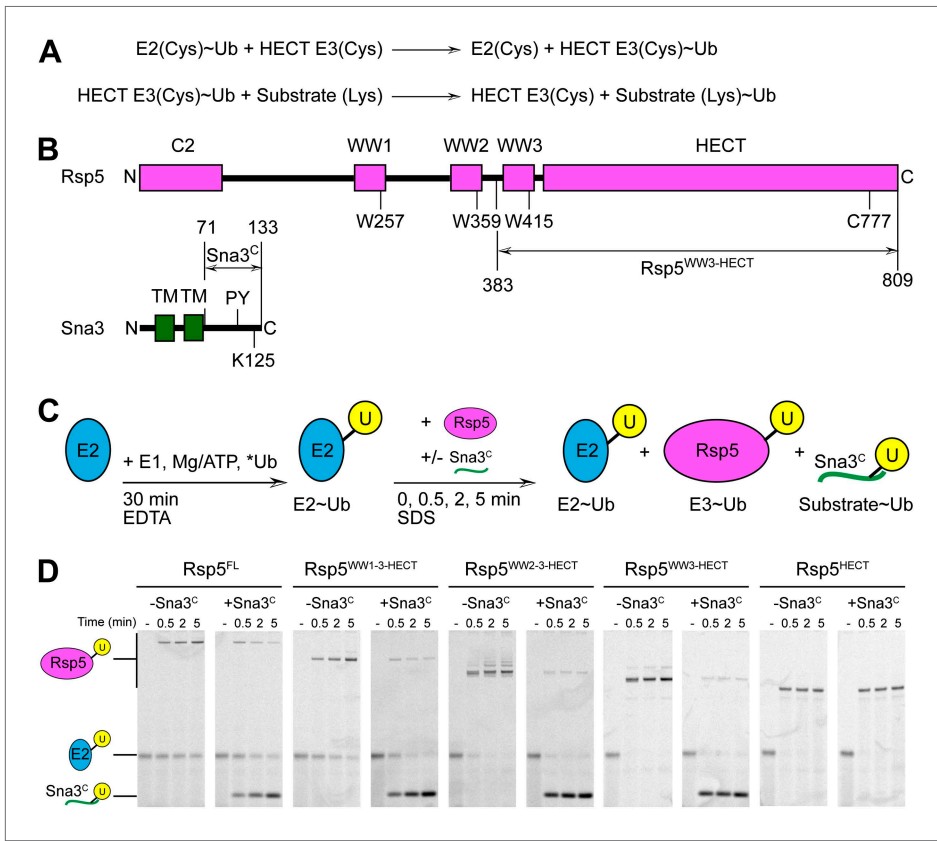

**Figure 1**. Rsp5$^{WW3-HECT}$-Sna3$^C$ as a minimal model HECT E3-substrate system to study Ub ligation. (**A**) Schematic view of two-step HECT E3 ubiquitination mechanism. First, Ub is transferred from an E2~Ub intermediate to the HECT E3 catalytic cysteine (Cys) to form a labile, thioester-linked E3~Ub intermediate. Second, Ub is transferred from the E3 Cys to a substrate's primary amino group, which is typically from a lysine side-chain. Ub can also be a substrate for the second reaction during polyubiquitination. (**B**) Schematic views of Rsp5 and Sna3 sequences. (**C**) Schematic description of pulse-chase assay. A thioester-bonded E2~Ub intermediate was enzymatically generated by mixing E1, the E2 UbcH5B, and a fluorescently labeled version of Ub for 30 min. This 'pulse' reaction was quenched by addition of EDTA. In the 'chase', wild-type or deletion mutant versions of Rsp5 were added alongside a synthetic peptide corresponding to Sna3$^C$. Formation of the Rsp5~Ub intermediate and the Sna3$^C$~Ub product were monitored at the indicated time-points. (**D**) Imaging of nonreducing SDS-PAGE gels monitoring pulse-chase fluorescent Ub transfer from E2 to Rsp5 to substrate for the indicated versions of Rsp5, in the absence or presence of Sna3$^C$. Rsp5$^{WW3-HECT}$ is the minimal version mediating Ub ligation to Sna3$^C$.

The following figure supplements are available for figure 1:

**Figure supplement 1**. Mutational data defining Rsp5$^{WW3-HECT}$ as a minimal E3 mediating Ub ligation to Sna3$^C$ and autoubiquitination.

fluorescently labeled version of Ub. After quenching formation of the E2~Ub intermediate, wild-type or truncation mutant versions of Rsp5 were added alongside a synthetic peptide corresponding to Sna3$^C$. Formation of the Rsp5~Ub intermediate and the Sna3$^C$~Ub product were monitored over time (*Figure 1C*). Thioester-linked intermediates were confirmed by their susceptibility to reduction by DTT (*Figure 1—figure supplement 1*). Complexes in which Ub was linked via an isopeptide bond from autoubiquitination of Rsp5 or ligation to a lysine on Sna3$^C$ were not susceptible to reduction. As expected, the isolated HECT domain forms a thioester-linked intermediate with Ub, but is incapable of promoting Ub ligation to substrate (*Figure 1D*, *Figure 1—figure supplement 1*). Inclusion of the third WW domain is required for Ub ligation to both Sna3$^C$ and also for autoubiquitination of Rsp5 itself (*Figure 1D*). Notably, little Rsp5$^{WW3-HECT}$ autoubiquitination is observed in the presence of the excess Sna3$^C$ substrate in our assays (*Figure 1D*).

## Alanine scanning mutagenesis of the Rsp5 HECT domain E3 suggests distinct architectures for E2-to-E3 Ub transfer and ligation

Interlobe flexibility observed upon comparing the structures of isolated HECT domains, and mutagenesis of residues linking the N- and C-lobes from a HECT domain, led to the idea that the N- and C-lobes adopt many different relative orientations during polyubiquitination (*Huang et al., 1999*; *Verdecia et al., 2003*; *Ogunjimi et al., 2005*). In this model, the catalytic requirements for ligation are encompassed exclusively in the C-lobe, its thioester-linked Ub, and the substrate. We considered that alternatively, tethering between the HECT domain N- and C-lobes could specify a limited set of distinct conformations for forming the thioester-linked E3~Ub intermediate and for ligation. To distinguish between these mechanisms, we performed comprehensive alanine mutagenesis of the HECT domain in Rsp5$^{WW3-HECT}$. We examined pulse-chase Ub ligation by 65 Rsp5$^{WW3-HECT}$ variants, each with one to four alanine mutations in the HECT domain (180 surface residues mutated in total) (*Figure 2A*, *Figure 2—figure supplement 1*). The rates of Ub transfer to Rsp5$^{WW3-HECT}$'s cysteine and subsequent ligation to the Sna3$^C$ peptide were too rapid to observe the thioester-linked E3~Ub intermediate at the earliest time-point in this assay. Thus, we considered that mutants displaying persistence of the E2~Ub intermediate would identify residues important for Ub transfer from E2 to the HECT domain. We anticipated that these mutations would map to the N- and C-lobe surfaces shown to bind E2~Ub in the prior crystal structure of a trapped E2~Ub-HECT complex (*Kamadurai et al., 2009*). By contrast, mutants displaying accumulation of the Rsp5$^{WW3-HECT}$~Ub intermediate would indicate defective ligation. We hypothesized that if the HECT domain adopts a specific architecture for ligation, then some mutations in the N-lobe may also hinder ligation. Alternatively, if the two HECT domain lobes are flexibly tethered during ligation, we expected that mutations disrupting Ub transfer to Sna3$^C$ would map to the C-lobe, where Ub is linked. Results from the Ala scan revealed that mutations defective in the first catalytic step of E2-to-E3 transfer all map to the E2~Ub binding site and the N-lobe/C-lobe interface identified in the previous E2~Ub-HECT domain structure (*Figure 2B*). By contrast, mutants that hindered Ub ligation to Sna3$^C$ map to distinctive surfaces on the N- and C-lobes (*Figure 2B*). Thus, the mutational data suggest a specific HECT domain architecture may be important for Ub ligation to Sna3$^C$, and this conformation would involve both the catalytic C-lobe and the distal N-lobe packing differently from the arrangement promoting ubiquitin transfer from E2-to-E3.

## Crystal structure of a trapped Rsp5$^{WW3-HECT}$ complex with Ub and Sna3$^C$ as a proxy for the catalytic intermediate

To visualize the Rsp5$^{WW3-HECT}$ architecture mediating ligation, we devised a method to bypass the instability of an Rsp5~Ub thioester bond, allowing generation of a stable proxy for a HECT E3~Ub-substrate intermediate. A synthetic Sna3$^C$ peptide with the acceptor Lys125 substituted with azidohomoalanine was linked by Click chemistry to a homobifunctional maleimide crosslinker (*Figure 3A,B*). One arm was crosslinked to Rsp5$^{WW3-HECT}$, and the other to a Cys side-chain substituted for Gly75 of Ub. This enabled us to determine the crystal structure of Rsp5$^{WW3-HECT}$ with the active site cysteine (Cys777) simultaneously crosslinked to the C-terminus of the Ub variant and to Sna3$^C$ (Rsp5$^{WW3-HECT}$xUbxSna3$^C$) (*Figure 3C*, *Figure 3—figure supplement 1*). The crystals contained two Rsp5$^{WW3-HECT}$xUbxSna3$^C$ complexes per asymmetric unit, which superimpose with ~0.5 Å RMSD. Accordingly, only one is discussed here. The individual domains in Rsp5$^{WW3-HECT}$xUbxSna3$^C$ overlay well with previous HECT structures. Noncovalent interactions between the HECT domain C-lobe and Ub involved in Ub transfer from E2-to-E3 (*Kamadurai et al., 2009*; *Maspero et al., 2013*) are preserved in the Rsp5$^{WW3-HECT}$xUbxSna3$^C$ structure.

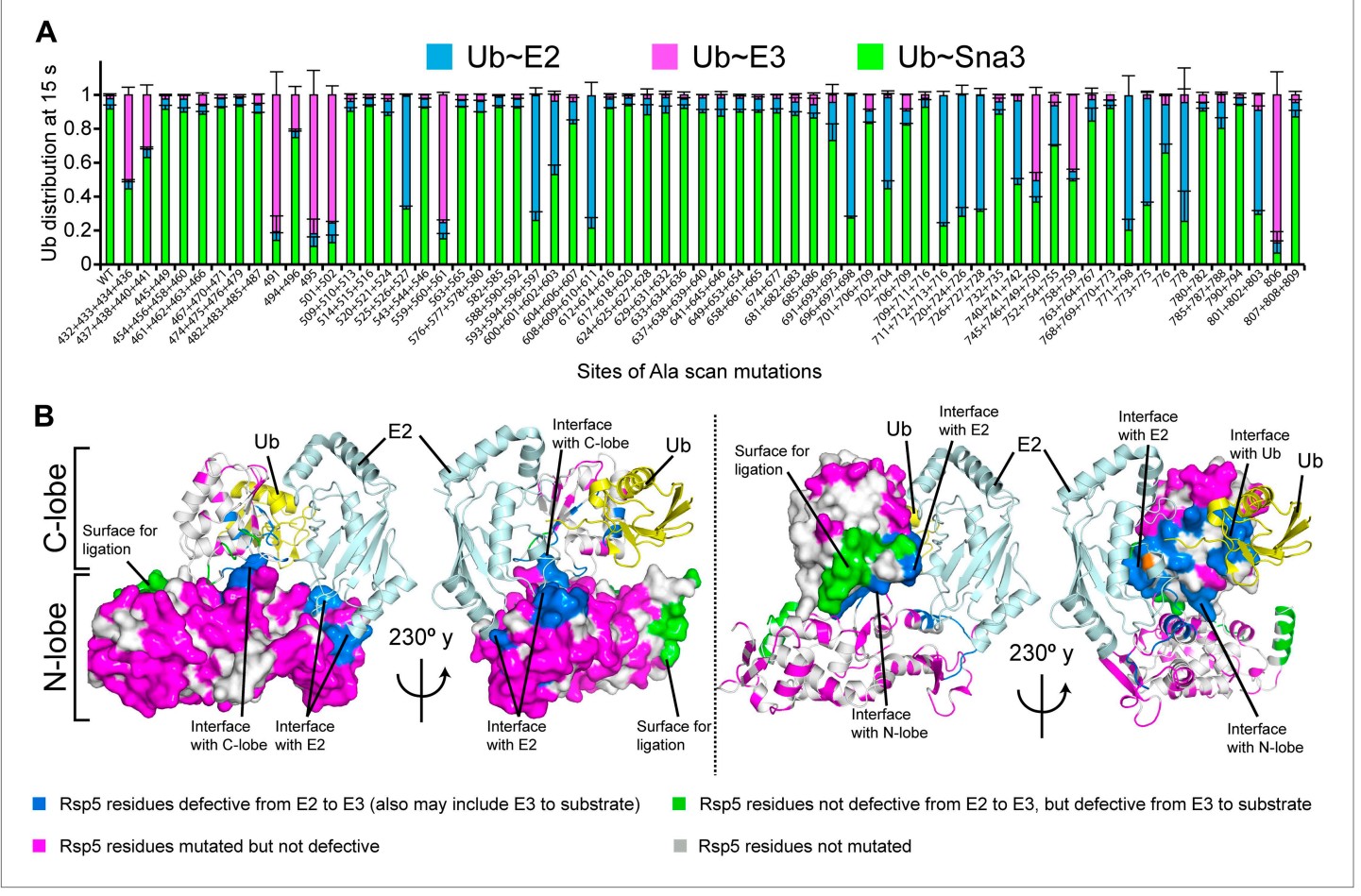

**Figure 2**. Alanine scanning mutagenesis suggests distinct specific HECT domain architectures to receive and ligate ubiquitin. (**A**) Summary of effects of indicated HECT domain Ala mutations on pulse-chase fluorescent Ub transfer from E2 to Rsp5[WW3-HECT] and then from Rsp5[WW3-HECT] to Sna3[C]. The assay scheme is shown in **Figure 1C**, except a 15 s chase was used. Cyan bars—accumulation of thioester-linked E2~Ub intermediate, reflecting a defect in Ub transfer from E2 to E3. Purple bars—accumulation of thioester-linked E3~Ub intermediate, reflecting a defect in Ub transfer from E3 to substrate. Green bars—ratio of Sna3[C]~Ub product formed. Standard deviations are calculated from three independent replications. (**B**) Locations of mutations hindering E2-to-Rsp5[WW3-HECT] (blue) or Rsp5[WW3-HECT]-to-substrate (green) Ub transfer mapped on prior E2~Ub-HECT domain structure (**Kamadurai et al., 2009**). Magenta surfaces represent residues mutated and found not essential for activity.

The following figure supplements are available for figure 2:

**Figure supplement 1**. Representative raw data from Ala scan.

The Sna3[C] PPXY motif binds Rsp5's WW3 domain in a typical manner (**Vijay-Kumar et al., 1987**; **Kanelis et al., 2001**; **Kim et al., 2011**). However, the overall Rsp5[WW3-HECT]xUbxSna3[C] assembly is distinctive, with the HECT domain active site and its covalently linked Ub facing the Sna3[C] substrate (**Figure 3C**). Importantly, the locations of the HECT domain Ala scan mutations that selectively impaired ligation map to surfaces that anchor the Rsp5 lobes and Ub in the Rsp5[WW3-HECT]xUbxSna3[C] structure (**Figure 3D**).

The HECT domain~Ub architecture is established by extensive interactions that bury ~2800 Å of surface area between the N- and C-lobes, and Ub. Below we describe how (1) the two lobes of the HECT domain cradle Ub's C-terminus to prime the thioester bond for the ligation reaction, (2) the HECT domain conformation is established by interactions between the C- and N-lobes, (3) the HECT domain architecture results in a composite Ub/N-lobe/C-lobe catalytic center, with critical N-lobe residues placed adjacent to Ub's thioester linkage to the catalytic Cys in the C-lobe, (4) the catalytic architecture projects the E3~Ub thioester bond toward the substrate and prioritizes substrate lysines for ubiquitination, and (5) together with the prior structure of a trapped E2~Ub-HECT intermediate

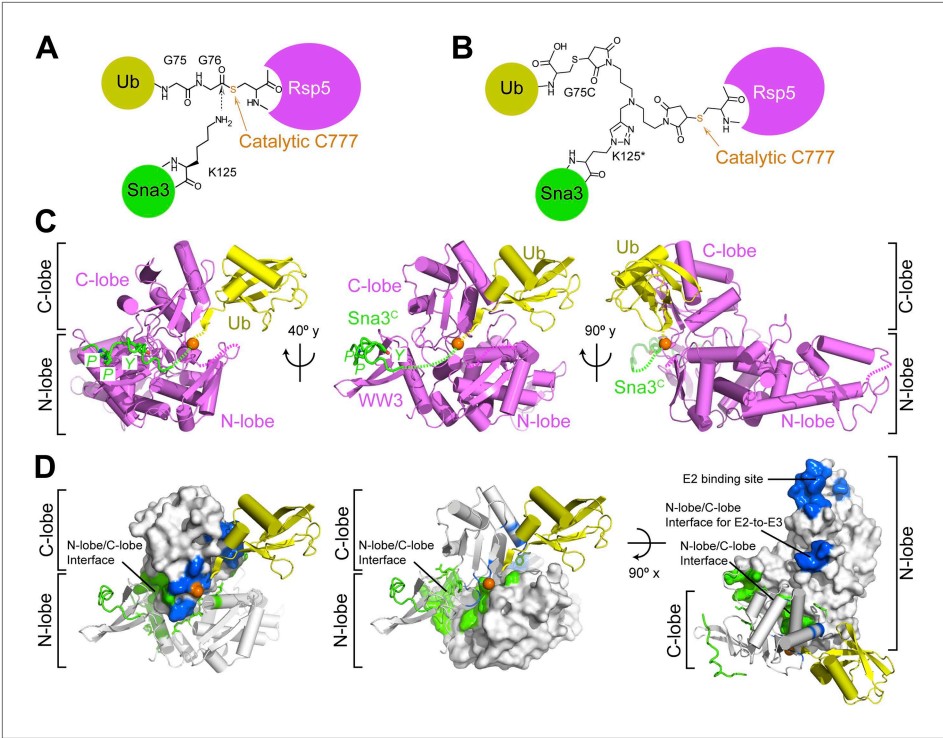

**Figure 3**. Structure of a trapped proxy for an Rsp5~Ub-Sna3$^C$ intermediate. (**A**) Chemical representation of substrate lysine attack of an Rsp5~Ub thioester bond. (**B**) Chemical representation of the trapped proxy, depicting covalent bonds linking Rsp5 Cys777, a Ub Cys75 (in place of Gly75-Gly76) and Sna3$^C$ Cys125 (in place of Lys125) in Rsp5$^{WW3-HECT}$xUbxSna3$^C$ complex. (**C**) Three views of Rsp5$^{WW3-HECT}$ (violet) xUb (yellow) xSna3$^C$ (green), with catalytic Cys shown as orange sphere. Regions not observed in electron density are indicated with dashed lines. (**D**) Locations of Ala scan mutations hindering E2-to-Rsp5$^{WW3-HECT}$ (blue) or Rsp5$^{WW3-HECT}$-to-substrate (green) Ub transfer mapped on Rsp5$^{WW3-HECT}$xUbxSna3$^C$ structure.

The following figure supplements are available for figure 3:

**Figure supplement 1**. Electron density for Rsp5$^{WW3-HECT}$xUbxSna3$^C$ structure.

(**Kamadurai et al., 2009**) the new data rationalize the necessity for rotation about the HECT domain N- and C-lobes by enabling visualization of a HECT E3 Ub transfer cascade.

## The Ub~HECT covalent linkage is oriented by Ub interactions with the HECT domain N- and C-lobes

The HECT domain architecture in the Rsp5$^{WW3-HECT}$xUbxSna3$^C$ structure results in Ub being sandwiched between the C- and N-lobes (**Figure 4A**). On one side, the interface between Ub's globular domain and the HECT C-lobe superimposes with the corresponding region of the E2~Ub-HECT structure (0.99 Å RMSD) (**Figure 4B**). Here, Ub's Ile36/Gln40/Leu71/Leu73 hydrophobic patch contacts Leu771, Leu798, Ala799, Glu802, and Thr803 of Rsp5 (**Figure 4C,D**). Accordingly, mutating these interface residues of Ub is lethal to yeast (**Sloper-Mould et al., 2001**; **Kamadurai et al., 2009**), and Ub transfer from E2 is impaired for L771A/L798A and E802A/T803A/I804A mutant versions of Rsp5$^{WW3-HECT}$ tested in our Ala scan, and for the L73A mutant versions of Ub (**Figures 2A and 4E**). The E3~Ub intermediate accumulates and the rate of Ub ligation to Sna3$^C$ is also slowed for these mutants, although a relatively lesser effect on Rsp5-mediated Ub ligation to substrate may reflect the larger interface stabilizing E3–Ub interactions for the second reaction step (**Figure 4E**). Neither Ub transfer from E2 to Rsp5$^{WW3-HECT}$ nor from Rsp5$^{WW3-HECT}$ to Sna3$^C$ is substantially affected by an Ala substitution for Ub's Ile44, which does not make contacts in either the prior E2~Ub–NEDD4L$^{HECT}$ or the current Rsp5$^{WW3-HECT}$xUbxSna3$^C$ structures (**Figure 4E**). Conversely, Ile44 is essential for RING E3-mediated Ub ligation and for E3-independent polyubiquitination by some E2s (**Saha et al., 2011**; **Wickliffe et al., 2011**; **Dou et al., 2012**; **Plechanovova et al., 2012**; **Pruneda et al., 2012**).

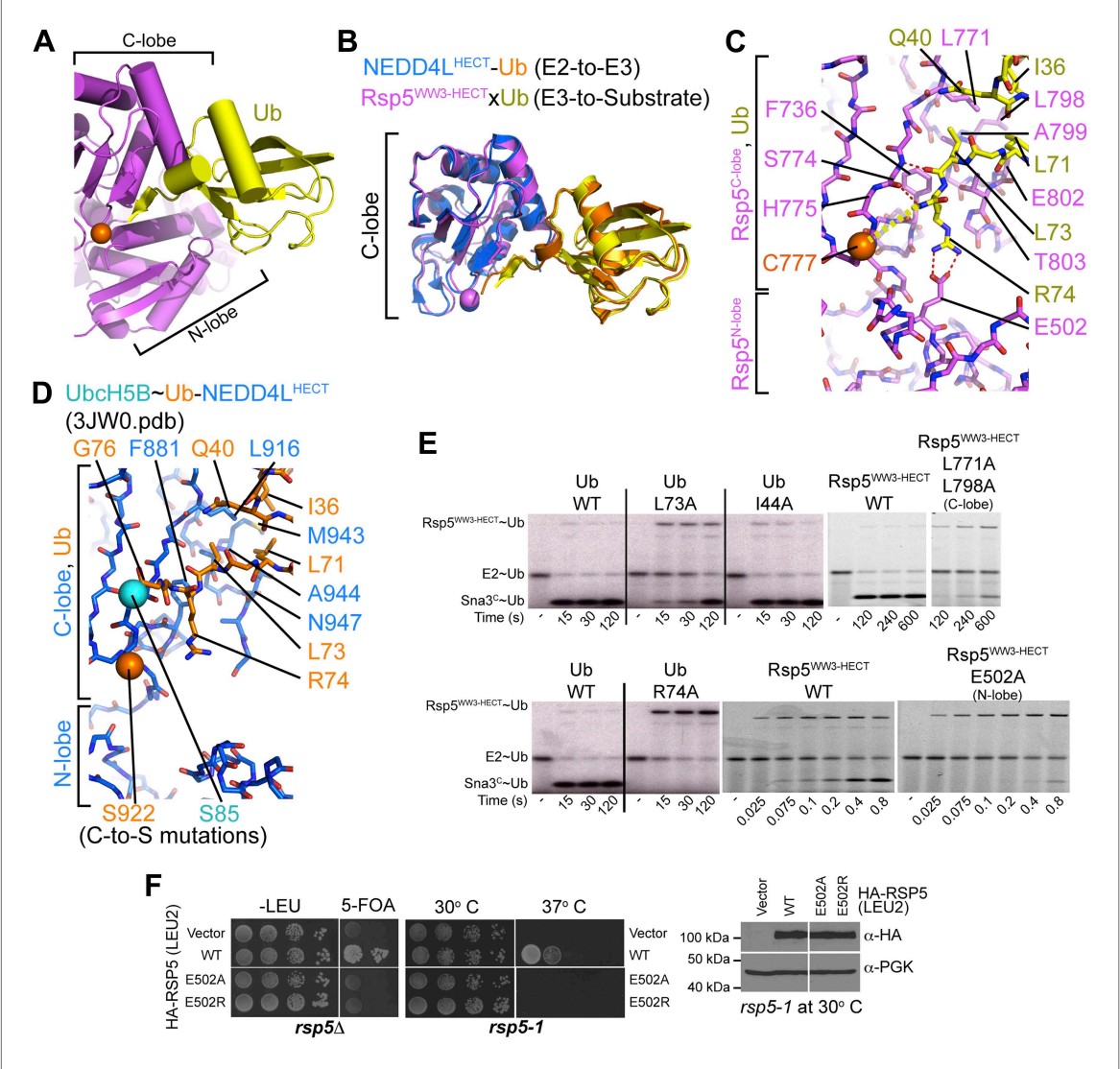

**Figure 4**. HECT domain–Ub interactions for ligation. (**A**) Ub's C-terminal tail and its covalent linkage to the Rsp5 catalytic Cys are shown sandwiched between the Rsp5 N- and C-lobes in the crystal structure of Rsp5^WW3-HECT (violet) xUb (yellow) xSna3^C (not shown). (**B**) C-lobe-Ub portions superimposed for Rsp5^WW3-HECT (violet) xUb (yellow) xSna3^C (not shown) and E2 (not shown)~Ub (orange)-NEDD4L^HECT (blue) (***Kamadurai et al., 2009***). (**C**) Close-up view of interactions between Rsp5 C-lobe and the C-terminal tail of covalently linked Ub in the crystal structure of Rsp5^WW3-HECT (violet) xUb (yellow) xSna3^C (not shown). (**D**) Close-up view of interactions between NEDD4L C-lobe and the C-terminal tail of the E2-linked Ub in the crystal structure of E2 (only the Cys-to-Ser mutation at the active site is shown) ~Ub (orange)-NEDD4L^HECT (blue) (***Kamadurai et al., 2009***). (**E**) Nonreducing gels from pulse-chase E2-to-Rsp5-to-Sna3^C Ub transfer assay, using indicated versions of Rsp5^WW3-HECT and of fluorescent or radioactive Ub. Bands corresponding to thioester-linked E2~Ub and Rsp5^WW3-HECT~Ub intermediates and isopeptide-bonded Sna3^C~Ub product are indicated. (**F**) Yeast complementation assays for the indicated HA-Rsp5 mutants.

The Ub C-terminal tail adopts an extended conformation, secured through an intermolecular β-sheet with the HECT β-strand leading to the covalent linkage with the catalytic Cys777 (***Figure 4C***). Here, backbone hydrogen bonds are formed between Ub residues 73 and 75 and Rsp5's Ser774. Because this three-residue β-sheet is not found when Ub is bound to E2 in the E2~Ub-HECT complex (***Kamadurai et al., 2009***), this structure apparently results from covalent linkage to the E3 Cys, and is likely specific for ligation (***Figure 4C,D***).

Three-way interactions between the HECT domain N-lobe, C-lobe, and the covalently linked Ub are anchored by Ub's Arg74 (***Figure 4C***). The aliphatic portion of Ub's Arg74 interacts with Phe736 from Rsp5's C-lobe. On the other face of Ub, the Arg74 guanidino group packs in a region of the N-lobe dominated by acidic residues, and directly contacts Glu502, which corresponds to a site of mutation

in the E6AP HECT E3 in Angelman syndrome (*Huang et al., 1999*; *Cooper et al., 2004*). Mutation of Arg74 in Ub or Glu502 in the Rsp5 HECT domain N-lobe impaired Ub ligation to Sna3$^C$ in vitro (*Figure 4E*), and Rsp5's essential function in vivo (*Figure 4F*) (*Sloper-Mould et al., 2001*). Through rapid quench-flow kinetic analyses, we measured a sevenfold defect in the rate of Ub transfer from Rsp5$^{WW3-HECT}$ to Sna3$^C$ for the E502A mutant in our assay (*Figure 5*).

## Interlobe interactions stabilize HECT domain conformation

In the Rsp5$^{WW3-HECT}$xUbxSna3$^C$ structure, almost half the N-lobe nestles and orients the C-lobe (*Figure 5A*). Notably, the importance of these N-lobe/C-lobe interactions is confirmed by impaired ligation activity for every Ala scan mutant mapping to this interface (*Figures 2A and 3D*). For example, Arg560 and

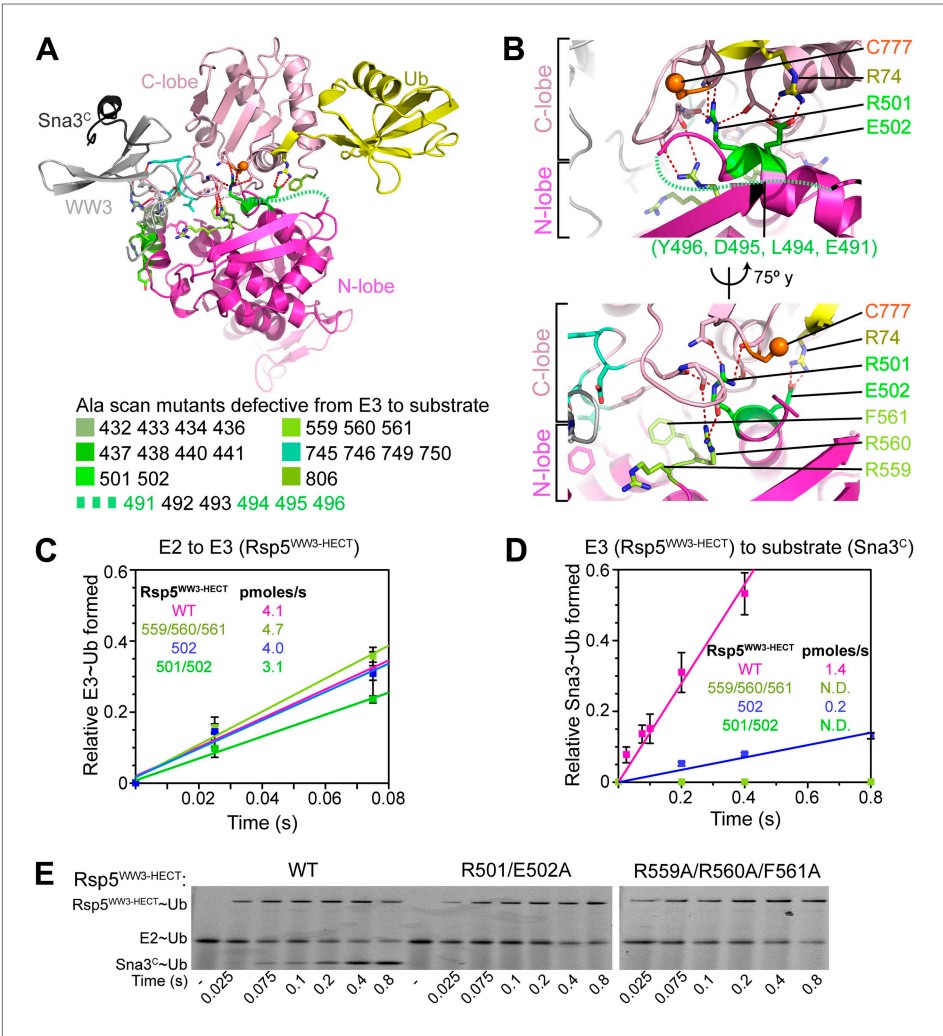

**Figure 5**. Ala mutations selectively impaired for ligation map to N-lobe/C-lobe interface in Rsp5$^{WW3-HECT}$xUbxSna3$^C$ crystal structure. (**A**) Rsp5$^{WW3-HECT}$xUbxSna3$^C$ crystal structure highlighting locations of Ala mutants (sticks, colored by mutant as indicated) selectively impaired for ligation. Dotted line indicates approximate locations of mutations in the N-lobe not visible in electron density. (**B**) Close-up views highlighting N-lobe/C-lobe interface locations of Ala mutants selectively impaired for ligation. Dotted line indicates approximate locations of mutations in the N-lobe not visible in electron density. (**C**) Initial rates of fluorescent Ub transfer from E2 to the indicated versions of Rsp5$^{WW3-HECT}$ measured in rapid quench-flow pulse-chase assays. (**D**) Initial rates of fluorescent Ub transfer from the indicated versions of Rsp5$^{WW3-HECT}$ to Sna3$^C$ measured in rapid quench-flow pulse-chase assays. (**E**) Fluorescent images of nonreducing SDS-PAGE gels showing rapid-quench flow pulse-chase fluorescent Ub transfer from E2 to Rsp5$^{WW3-HECT}$ to Sna3$^C$, for the indicated versions of Rsp5$^{WW3-HECT}$. Bands corresponding to thioester-linked E2~Ub and Rsp5$^{WW3-HECT}$~Ub intermediates and isopeptide-bonded Sna3$^C$~Ub product are indicated.

Phe561 from the N-lobe map to the core of the interface with the C-lobe (*Figure 5B*). Accordingly, the Arg559, Arg560, Phe561 triple Ala mutant was severely impaired in Ub ligation to Sna3$^C$, with no defect in the rate of E2-to-Rsp5 Ub transfer in our assay (*Figure 5C–E*). Similarly, Arg501 from the N-lobe further anchors the C-lobe, and the combination of R501A and E502A mutations completely abrogates ligation in our assay, with a more severe defect than E502A alone (*Figure 5B–E*). There are lesser effects of mutating side-chains in the Lys432/Arg433/Asp434/Arg436 and Arg437/Lys438/Ile440/Tyr441 combinations in the N-lobe, and the Val745/Asn746/Lys749/Asp750 combination in the C-lobe, presumably because of their more peripheral locations in the interdomain interface (*Figure 5A*).

Furthermore, the C-lobe's Phe806, also known as the '-4 phenylalanine' required for the polyubiquitination cycle (*Salvat et al., 2004*), loosely approaches Phe505 and Leu506 in the N-lobe (*Figure 6A*). The F806A mutant was severely impaired in the Ala scan, and pulse-chase assays show specific defects in Ub ligation upon mutating either Rsp5$^{WW3-HECT}$'s Phe806 to Leu, or the Phe505 and Leu506 from the

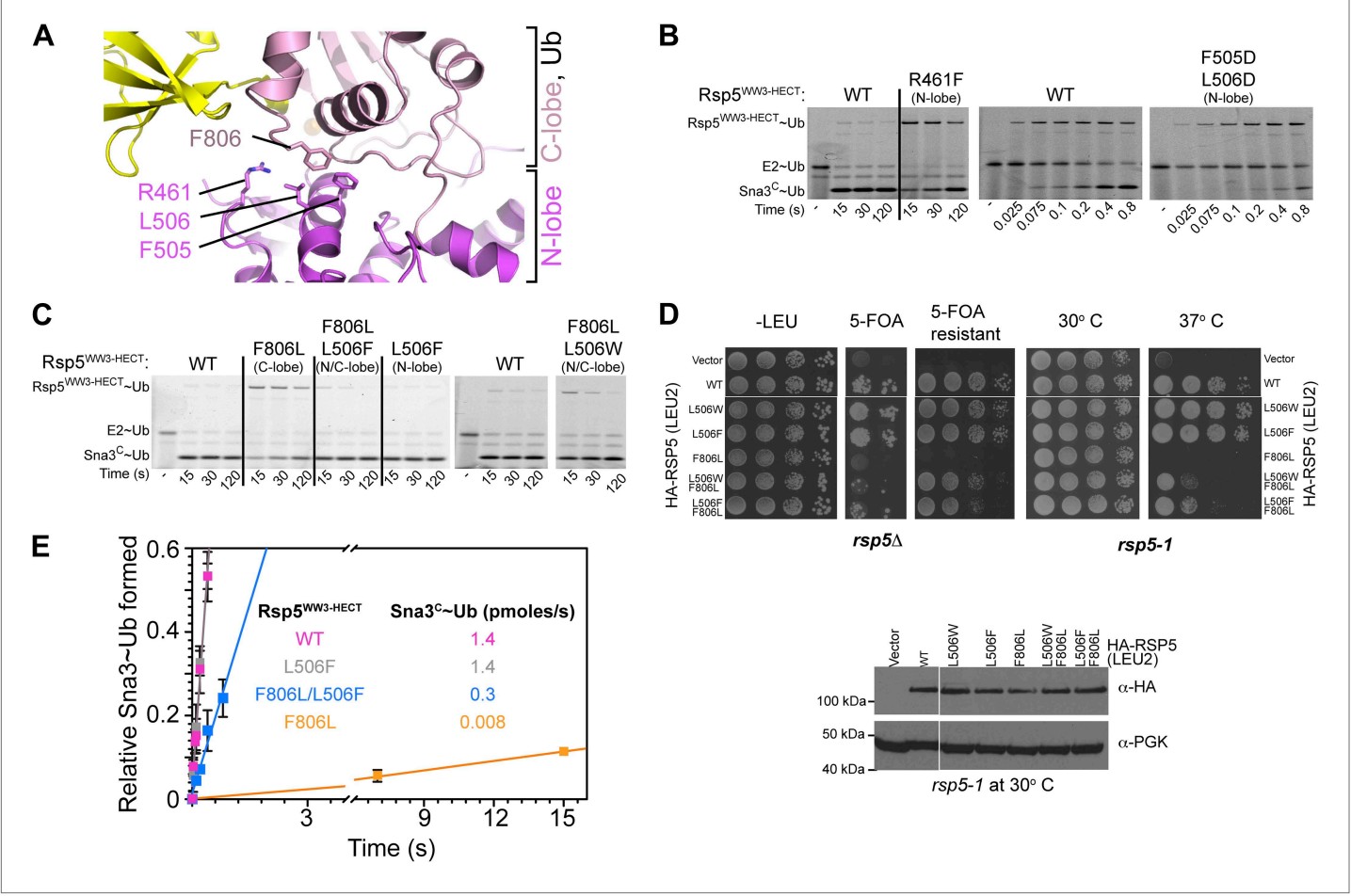

**Figure 6**. HECT domain C-lobe/N-lobe interactions anchoring architecture for ligation. (**A**) Close-up view of a portion of the C-lobe/N-lobe interface in Rsp5$^{WW3-HECT}$xUbxSna3$^C$ structure. (**B,C**) Nonreducing gels from pulse-chase transfer assay of fluorescent Ub from E2-to-Rsp5-to-Sna3$^C$ using indicated mutants of Rsp5. Bands corresponding to thioester-linked E2~Ub and Rsp5$^{WW3-HECT}$~Ub intermediates and isopeptide-bonded Sna3$^C$~Ub product are indicated. (**D**) Left: HA-tagged WT and mutant *rsp5* alleles housed on low copy plasmids were assessed for their ability to complement the essential function of *RSP5* in either serially diluted *rsp5-1* temperature-sensitive cells grown at restrictive 37°C or in *rsp5Δ* null cells after eviction of wild-type *RSP5* plasmid on 5-FOA. Below: whole cell lysates of *rsp5-1* transformants immunoblotted for HA and PGK. (**E**) Rates of pulse-chase fluorescent Ub ligation from the indicated versions of Rsp5$^{WW3-HECT}$ to Sna3$^C$.

The following figure supplements are available for figure 6:

**Figure supplement 1**. A role for HECT domain C-terminus in ligation.

N-lobe simultaneously to aspartates (*Figures 2A and 6B*). Leu506 is adjacent to Arg461, and a bulky R461F mutation also hinders ligation (*Figure 6B*). In agreement with the structural data, these mutations are ligation-specific, and do not impact formation of the thioester-linked Rsp5<sup>WW3-HECT</sup>~Ub intermediate. We devised an experiment to dissect the role of the -4 Phe during ligation, by complementing the defect of the C-lobe F806L with a compensatory aromatic residue at position 506 of the N-lobe. Indeed, the defect from the F806L mutation in the C-lobe is largely ameliorated by complementary N-lobe L506W or L506F mutations both in vivo and in vitro (*Figure 6A,C–E*). Moreover, rapid quench-flow kinetic analyses confirmed wild-type rates of E2-to-E3 Ub transfer for these mutants, and also observed 43-fold rescue in the rate of Ub ligation to Sna3$^C$ when the defective F806L mutant was combined with L506F mutation (*Figure 6E*). This compensation is selective, as L506F does not influence the deleterious effects of mutating the structurally distal residue Phe778, even after extended reaction times. Overall, these data demonstrate that an aromatic side-chain has a critical function in the ligation reaction by anchoring the two HECT domain lobes.

## A functional role for Rsp5's C-terminus

Because Phe806 is near to the C-terminus, we considered whether the Rsp5 C-terminal sequence may play a role in ligation. Deleting Rsp5's C-terminal residue severely impairs Ub ligation to Sna3$^C$, although not to the same extent as the extremely deleterious D495A mutant identified in our Ala scan (*Figure 6—figure supplement 1A,B*). Our results are consistent with the recent finding that deleting Rsp5's C-terminal residue also impairs substrate-independent polyubiquitination by the isolated HECT domain from Rsp5 (*Maspero et al., 2013*). However, although it has been suggested that a C-terminal acidic side-chain may perform a catalytic role in the ligation reaction (*Maspero et al., 2013*), we observed no defect upon simultaneously mutating all three of Rsp5's C-terminal residues to alanines (*Figure 2A*), or mutating the Rsp5's C-terminal residue to Arg (*Figure 6—figure supplement 1A*). Future studies will be required to definitively determine the role of the C-terminus, which has not been observed in any structure of a HECT E3, including Rsp5<sup>WW3-HECT</sup>xUbxSna3$^C$. However, we speculate that for Rsp5 and possibly other HECT E3s, the C-terminus itself may contribute to the ligation reaction. Indeed, C-terminal epitope tags have been shown to hinder activity (*Salvat et al., 2004*). Alternatively, the C-terminus may stabilize the conformation of the HECT~Ub complex. In this regard, we note that in the Rsp5<sup>WW3-HECT</sup>xUbxSna3$^C$ structure, the backbone amides from Leu73 and Arg74 and other portions of Ub's extended C-terminal tail are partially exposed and could potentially interact with the acidic C-terminus of the HECT domain (*Figure 6—figure supplement 1C*).

## A composite HECT domain catalytic center involving both the HECT C- and N-lobes

To generate models for the substrate lysine and a functionally important N-lobe loop not visible in the Rsp5<sup>WW3-HECT</sup>xUbxSna3$^C$ electron density, we used the structure prediction program Rosetta. First, we determined the orientation of the thioesterified ubiquitin tail using constraints on the ubiquitin locations derived from the crystal structure. To accommodate the ubiquitin location, the catalytic cysteine must adopt the gauche+ rotamer, which allows formation of the thioester and its packing against His775, Thr776, and the noncovalent interactions between Ub and the Rsp5 C-lobe (*Figure 7A*). Given the geometric requirements for isopeptide bond formation, this rotameric preference directs the acceptor lysine's path of attack and all models predicted that Lys125 of Sna3 packs against Phe778 (*Figure 7B*). His775, Thr776, and Phe778 correspond to residues that also interact with E2~Ub during formation of the HECT E3~Ub intermediate (*Kamadurai et al., 2009*). Their interaction with Ub was recently also observed in the structure of a HECT E3~Ub intermediate published while our manuscript was under consideration (*Maspero et al., 2013*). Furthermore, H775A and T776A mutations result in modest defects in Ub ligation to Sna3$^C$, and an F778A mutation results in a more severe defect (*Figure 7C*). While all the models demonstrate a similar approach for Sna3 Lys125, modeling the additional residues between this Lys125 and the PPXY motif revealed multiple orientations that could accommodate a proper Lys125 active-site approach. Accordingly, neither Ala nor Glu substitutions adjacent to the acceptor lysine influenced substrate selection (*Figure 7—figure supplement 1*).

Although the Rsp5 loop comprising residues 491–495 is not visible in the structure, our Ala scan results revealed that Glu491 and Asp495 are essential for the ligation reaction, and defective ligation was also observed for the L494A Y496A double mutant (*Figure 2A*). Accordingly, E491A and D495A Rsp5 mutants are also defective in supporting yeast growth (*Figure 7D*). To gain initial insights

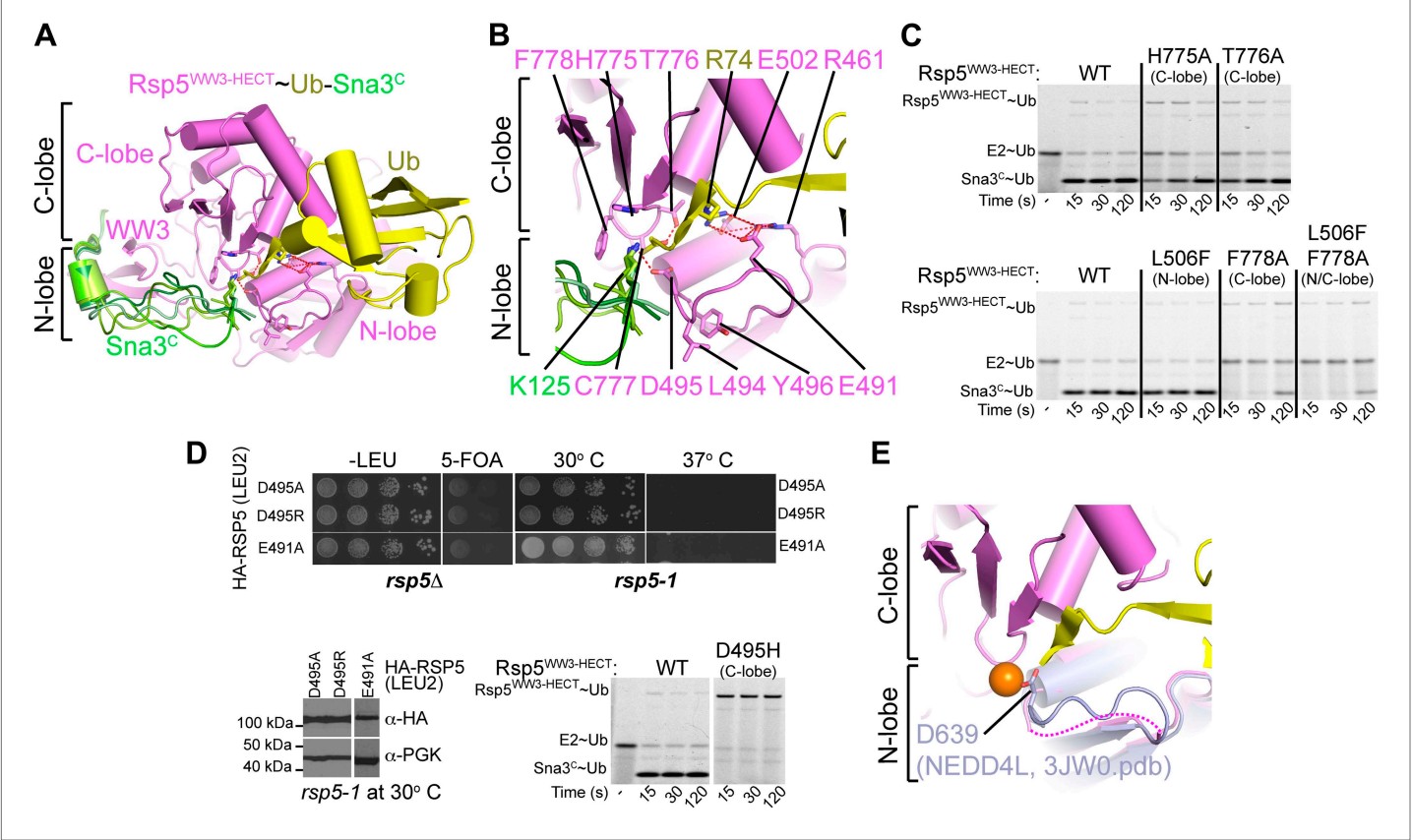

**Figure 7**. A distinctive active site for HECT E3-mediated ligation. (**A**) Rosetta-generated models for Sna3$^C$ (different models in different shades of green) target lysine approaching Rsp5$^{WW3-HECT}$ (violet) active site Cys777 thioester-linked to Ub (yellow). (**B**) Model for residues aligning thioester-bound Ub and Sna3$^C$ acceptor lysine approaching the active site. (**C**) Nonreducing gels from pulse-chase assay for transfer of fluorescent Ub from E2-to-Rsp5-to-Sna3$^C$ using indicated versions of Rsp5$^{WW3-HECT}$. Bands corresponding to thioester-linked E2~Ub and Rsp5$^{WW3-HECT}$~Ub intermediates and isopeptide-bonded Sna3$^C$~Ub product are indicated. (**D**) Yeast complementation assays for the indicated HA-Rsp5 mutants. (**E**) Close-up of structural superposition of N-lobes showing catalytic loop from NEDD4L$^{HECT}$ (**Kamadurai et al., 2009**) not visible in Rsp5$^{WW3-HECT}$ (violet) structure.

The following figure supplements are available for figure 7:

**Figure supplement 1**. Residues proximal to the acceptor lysine in Sna3 play insignificant roles in ubiquitination.

into potential functions of this region, we superimposed the structure of the HECT domain N-lobe from NEDD4L (**Kamadurai et al., 2009**) onto Rsp5$^{WW3-HECT}$xUbxSna3$^C$ (**Figure 7E**). This suggested that the 491–495 loop from the N-lobe is located adjacent to the thioester bond between Ub and the catalytic Cys in the C-lobe, and provides a rationale for the importance of the HECT domain architecture for the ligation reaction. Structural modeling of this critical N-lobe loop using Rosetta suggests that Glu491 could contribute to positioning the loop and may participate in an electrostatic network involving Arg74 of Ub and Arg461 of Rsp5 (**Figure 7B**). Furthermore, these structural models demonstrate that the acidic loop can adopt several permissive orientations that allow the carboxyl group of Asp495 to approach the epsilon nitrogen of the acceptor lysine (**Figure 7B**). Asp495 may activate ubiquitin ligation through a variety of mechanisms including guiding the lysine into the active site or by contributing, either directly or indirectly, to the deprotonation of the substrate lysine. Direct deprotonation may be less likely as we were unable to rescue catalytic activity either with a D495H mutant, or chemically by adding high concentrations of formate or acetate (**Figure 7D** and data not shown).

## General importance of architecture for HECT E3-mediated Ub ligation

We wished to address whether the catalytic conformation is specific for Sna3$^C$ and Rsp5, or whether there is a general NEDD4-family HECT E3 architecture for ligation. As a first test, we sought to

examine the effects of alanine substitutions on another Rsp5 substrate. As NEDD4-family HECT E3s can transfer Ub to a free Ub acceptor, we tested the effects of mutations on di-Ub synthesis. There is a striking similarity in the effects of the Ala mutations in pulse-chase assays for Ub ligation to Sna3[C] and to free Ub as the substrate (*Figure 8*). Some subtle differences appear due to the high concentration of free Ub ameliorating the deleterious effects of a few mutations on E2-to-E3 Ub transfer, potentially due to Ub's noncovalent association with the HECT domain N-lobe stabilizing the E2 binding site. Also, one mutant (L494A, Y496A) showed a greater defect in di-Ub synthesis, perhaps marking the site where Ub binds as a substrate. Nonetheless, the overall agreement between mutational effects on modification of the two distinctive substrates suggests that a specific HECT domain architecture is generally important for Rsp5-mediated Ub ligation.

As a first step toward testing whether other HECT E3s adopt a similar architecture for ligation, we examined whether other NEDD4-family HECT E3s showed defects upon mutating their N-lobe aspartate corresponding to Rsp5's Asp495, which is distal from the catalytic center for Ub transfer from E2 to NEDD4-family E3s (*Kamadurai et al., 2009*), but maps to the composite active site for ligation (*Figure 9*). As with the Rsp5[WW1-3-HECT] D495A mutant, the corresponding D584A and D639A mutant versions of NEDD4[WW1-4-HECT] and NEDD4L[WW1,3-4-HECT] form an E3~Ub thioester intermediate. However, the mutations completely eliminate autoubiquitination. Taken together, the results raise the possibility that NEDD4-family HECT E3s use a common, specific catalytic architecture for Ub ligation to various substrates.

### The catalytic architecture provides a mechanism for target lysine selection

A fundamental question is how substrate lysines are selected for ubiquitination. A mechanism for prioritizing lysines for modification is suggested by the ligation-primed architecture of the

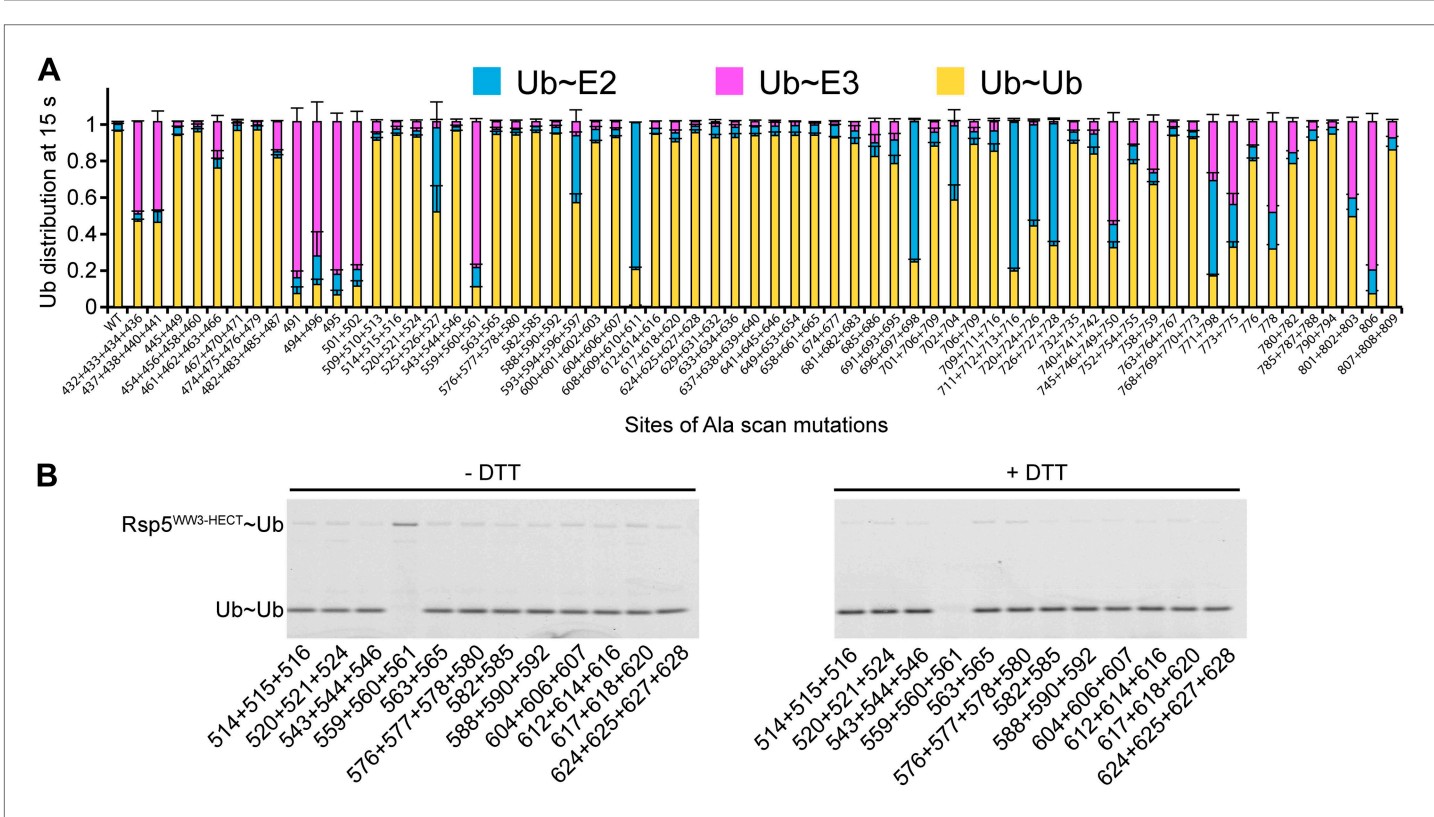

**Figure 8**. Ala scan for HECT domain surfaces required for di-Ub synthesis. (**A**) Summary of the effects of indicated Rsp5[WW3-HECT] HECT domain Ala mutations on pulse-chase fluorescent Ub transfer from E2 to Rsp5[WW3-HECT] and then to Ub. Total activity within each gel lane was determined by adding all intensities for all three species (E2~Ub, Rsp5[WW3-HECT]~Ub, and Ub~Ub) and was used to estimate the relative yield of each with respect to wild-type proteins. The error bar shows the standard deviation (SD) for three independent replicates. (**B**) Fluorescent imaging of representative nonreducing (left) and reducing (right) gels from assays used for data in **A**, monitoring pulse-chase fluorescent Ub transfer from E2 to the indicated versions of Rsp5[WW3-HECT] to Ub. The chase was 15 s.

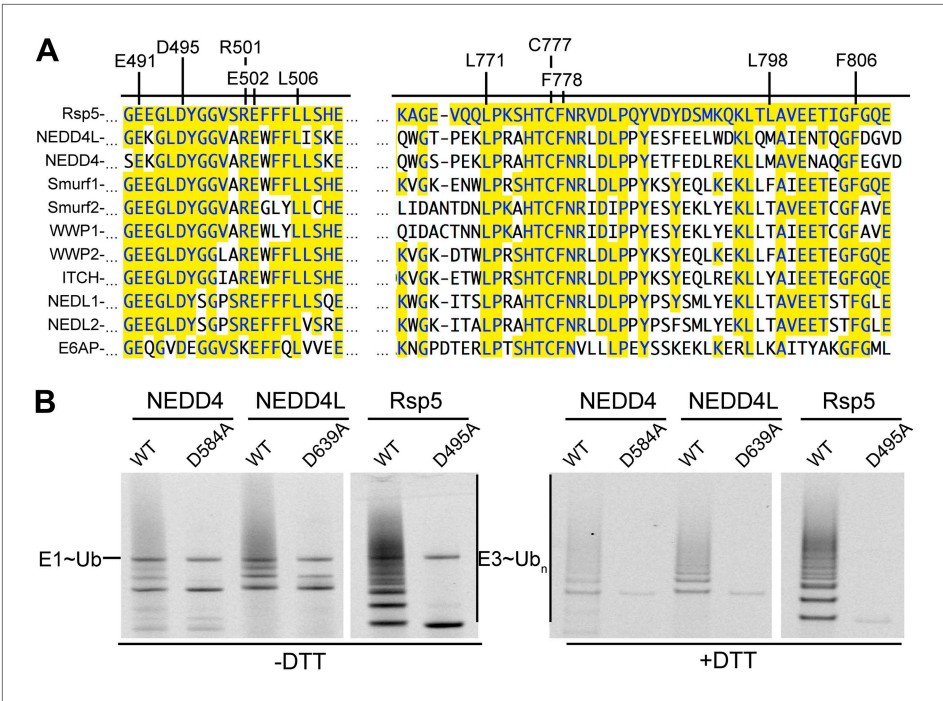

**Figure 9**. Key residues for ligation are conserved across HECT E3s. (**A**) Alignment of portions of sequence from Rsp5 with corresponding regions of human NEDD4L, NEDD4, Smurf1, Smurf2, WWP1, WWP2, ITCH, NEDL1, NEDL2, and E6AP showing conservation of key residues establishing the ligation mechanism. Identity to Rsp5 sequence is highlighted in yellow. (**B**) Fluorescent images of nonreducing (left) and reducing (right) gels of multiple turnover autoubiquitination assays for wild-type and the indicated Asp-to-Ala mutant versions of NEDD4[WW1-4-HECT], NEDD4L[1-3-4-HECT], and Rsp5[WW1-3-HECT].

Rsp5[WW3-HECT]xUbxSna3[C] complex. In the structure, Ub's C-terminus and the E3 catalytic Cys face the C-terminus of the substrate's PPXY motif anchored to the WW3 domain. However, the N-terminus of the bound Sna3[C] substrate is distal, and points away from the HECT domain active site.

We considered that the relative positions of the substrate-binding WW3 domain and the HECT domain N- and C-lobes might influence sites of Ub modification. In the structure, ~25 Å separates the HECT domain active site and the Sna3[C] PPXY motif. The distance predicts a requirement for a 10-residue spacer between the PPXY motif that binds the WW3 domain and the acceptor lysine that receives Ub from the HECT cysteine (*Figure 10*). By contrast, according to prior models suggesting that the domains are flexibly tethered during ligation, it would be possible to juxtapose the catalytic cysteine in the HECT domain C-lobe with a substrate lysine only two residues C-terminal of the WW3-bound PPXY motif in Sna3[C] (*Vijay-Kumar et al., 1987*; *Kanelis et al., 2001*; *Kim et al., 2011*). Thus, as a further test of the relevance of the fixed HECT domain architecture for ligation, we developed an assay for target lysine selectivity, using a TEV-cleavable HisMBP-fusion to Sna3[C] as a substrate. Fluorescent Ub~E2 was added to a mixture of Rsp5[WW1-3-HECT] and HisMBP-TEV-Sna3[C], and Ub transfer was terminated by adding DTT to reduce thioester-linked intermediates. After treatment with TEV protease, the levels of fluorescent Ub-ligated Sna3[C], HisMBP, and autoubiquitinated Rsp5[WW1-3-HECT] were compared (*Figure 10B*, *Figure 10—figure supplement 1*). For wild-type HisMBP-TEV-Sna3[C], Ub is predominantly ligated to Sna3[C]. Without HisMBP-TEV-Sna3[C], or with a mutant disrupting the HisMBP-TEV-Sna3[C] PPXY motif, Ub is ligated to Rsp5[WW1-3-HECT] itself. Conversely, if a HisMBP-TEV-Sna3[C] with a K125A mutation that eliminates Sna3 ubiquitination but still retains its Rsp5-binding PPXY motif is used, Ub is ligated to both HisMBP and Rsp5[WW1-3-HECT]. Therefore, the distribution of Ub-ligated Sna3[C], HisMBP, and Rsp5[WW1-3-HECT] products distinguishes preferences among different Ub-accepting lysines on substrates recruited via the same PPXY motif (Sna3[C] vs HisMBP), and on Rsp5[WW1-3-HECT] autoubiquitination as an internal control for ligation activity. To test whether the relative position of a lysine influences its ubiquitination, we performed assays with insertion and deletion mutants in the 15-residue linker between the PPXY motif and Ub-acceptor lysine of Sna3[C]. While substrates were

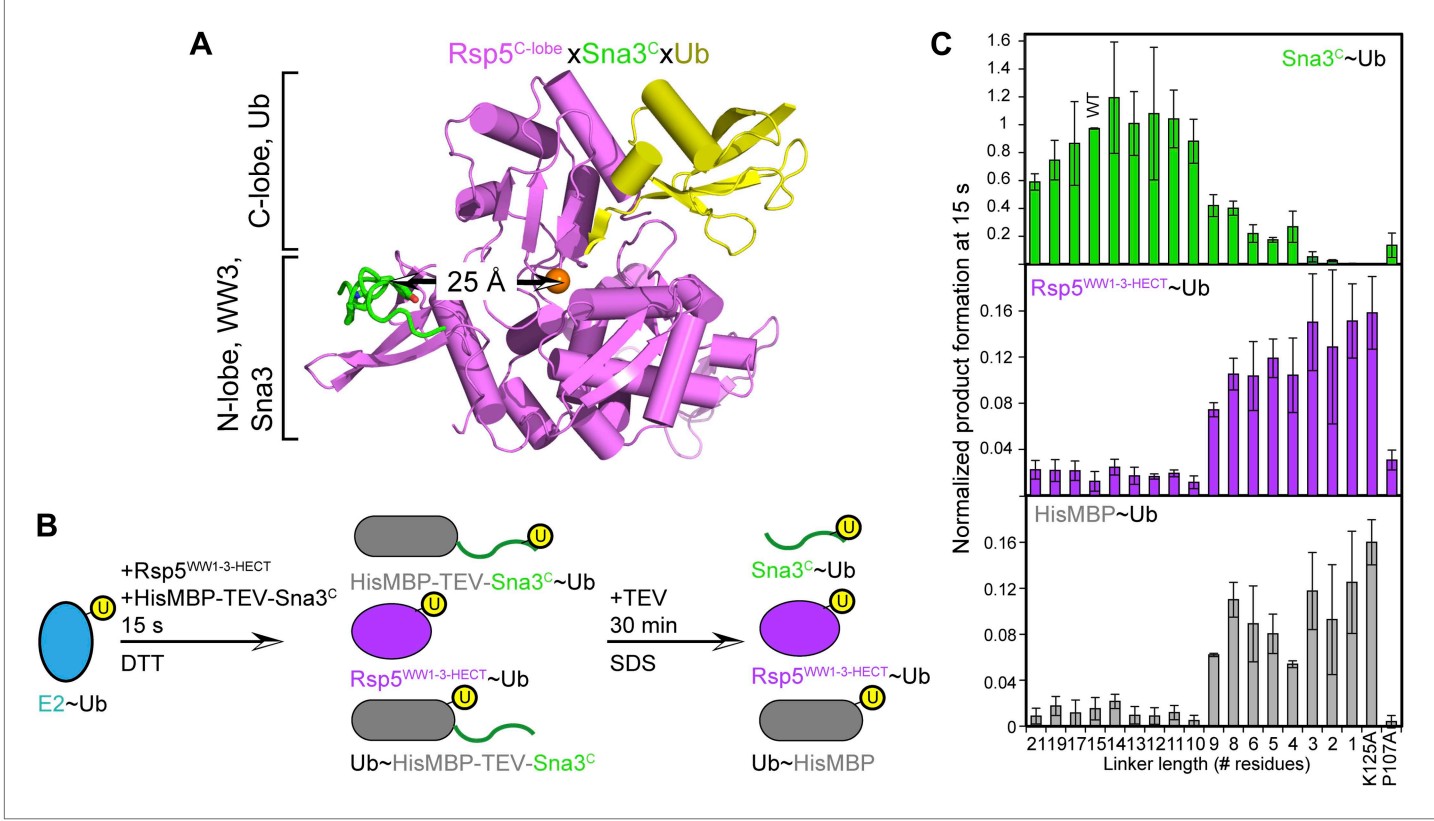

**Figure 10**. HECT domain architecture for ubiquitin ligation suggests mechanism for substrate lysine prioritization. (**A**) Structure of Rsp5[WW3-HECT] (violet) xUb (yellow) xSna3[C] (green), highlighting 25 Å distance between the alpha carbon of Sna3[C] Tyr109 in the PPXY motif and the sulfur of Rsp5[WW3-HECT] Cys777. (**B**) Schematic view of substrate selection assay. (**C**) Rsp5[WW1-3-HECT] selection between substrates with relative distributions of non-reducible Sna3[C]~Ub (green bar graph), autoubiquitinated Rsp5[WW1-3-HECT]~Ub (violet bar graph), and HisMBP~Ub (gray bar graph) products of reactions with substrates bearing the indicated number of residues between the Sna3[C] PPXY motif and lysine. Effects of Ala mutations in place of the Sna3[C] acceptor Lys125 and the WW3-binding Pro107 are shown as controls.

The following figure supplements are available for figure 10:

**Figure supplement 1**. Data supporting Rsp5[WW3-HECT] target Lys prioritization.

generally tolerant of insertions, reducing the linker to nine residues, irrespective of the sequence location, dramatically decreased Sna3[C] ubiquitination while increasing ubiquitination of HisMBP and Rsp5[WW1-3-HECT]. The strong spatial preference for target lysines suggests that the Rsp5 domains are not freely rotating during ligation. Furthermore, the sharp decline in ubiquitination upon reducing the number of residues linking the Ub acceptor lysine below 10 strongly correlates with the distance between the C-terminus of the Sna3[C] PPXY motif and the Rsp5 active site in the Rsp5[WW3-HECT]xUbxSna3[C] structure. Importantly, the N-terminally fused HisMBP was inefficiently ubiquitinated, despite containing 36 lysines. Nor was a lysine placed N-terminal to the PPXY motif. Overall, poor ubiquitination correlated with the structural location of the N-terminus of the PPXY motif, which projects away from the active site, although there may also be some context dependence to substrate selection based on the low activity toward an L121K K125A mutant (***Figure 10C***, ***Figure 10—figure supplement 1***). These data are also consistent with in vivo data showing that K125 is the major ubiquitination site of Sna3 and a Sna3-GFP fusion protein, with the GFP moiety undergoing a lower level of ubiquitination via its 20 lysines (***Stawiecka-Mirota et al., 2007***). Notably, the minimum required linker also correlates with the 12-14 residues between the LPXY motif and Ub acceptor lysines in the Rsp5 substrate Mga2p120 (***Bhattacharya et al., 2009***). Overall, the data indicate that the HECT E3 architecture primed for ligation prioritizes potential target lysines by their placement relative to a composite catalytic center for ubiquitination.

## Discussion

Taken together with a prior crystal structure of an E2~Ub-HECT domain complex (*Kamadurai et al., 2009*), our data allow visualization of a HECT E3's multistep Ub transfer cascade. Our systematic Ala scanning mutagenesis (*Figure 2*) confirms that the previous structure of a complex between oxyester-linked E2~Ub and a HECT domain (*Kamadurai et al., 2009*) explains how Ub is transferred from E2 to NEDD4-family HECT E3s, wherein the HECT domain C-lobe and catalytic cysteine face the E2. This position is dictated by noncovalent interactions between the HECT domain N-lobe and the E2, and between HECT domain C-lobe and the E2-linked Ub (*Kamadurai et al., 2009*). Our new data and a study published during manuscript revision (*Maspero et al., 2013*) suggest that after the E3~Ub intermediate is formed, the C-lobe remains associated with its thioester-linked Ub.

Here, to our knowledge, we provide the first snapshot of a HECT E3 poised to transfer Ub, enabled through our development of a technology for site-specific three-way crosslinking. The crystal structure of Rsp5 with its active site Cys simultaneously crosslinked to Ub's C-terminus and to a substrate reveals that the HECT domain C-lobe and Ub together rotate ~130° to face the substrate bound to an N-terminal WW domain (*Figure 11*). The three-way anchored HECT domain N-lobe/C-lobe~Ub architecture would reduce inherent conformational flexibility (*Verdecia et al., 2003*), and orient the E3~Ub intermediate toward the substrate for the ligation reaction (*Figure 3*). Furthermore, the fixed architecture for the HECT domain and its thioester-bound Ub during ligation would restrict orientations available to the WW domain, thus limiting the positions available for a lysine to access the active site.

We anticipate that other E3s functioning via a catalytic cysteine, including several effector proteins from bacterial pathogens and members of the RING-IBR-RING family (*Zhang et al., 2006*; *Rohde et al., 2007*; *Wenzel et al., 2011*), also activate Ub ligation through parallel mechanisms that align their E3~Ub intermediates. This theme is reflected by how mechanistically unrelated E3s (e.g., a SUMO E3 in the 'neither-HECT-nor-RING' class, and Ub E3s in the RING family) bind both E2 and its thioester-linked Ub-like protein (Ubl) to reduce conformational heterogeneity and optimally orient the E2~Ubl nucleophilic attack (*Reverter and Lima, 2005*; *Pruneda et al., 2011*; *Dou et al., 2012*; *Plechanovova et al., 2012*; *Pruneda et al., 2012*). Nonetheless, there are notable mechanistic differences between HECT E3s. For example, prior to the ligation reaction, there are differences during Ub transfer from the E2. For HECT E3s, orientation of the E2~Ub thioester bond involves extensive interactions between Ub and its 'target'—the C-lobe to which Ub will be transferred. By contrast, RING domains apparently orient the E2~Ub thioester bond for nucleophilic attack even in the absence of any downstream acceptor. As a result, RING E3s promote discharge of associated E2~Ub thioester intermediates, whereas E2~Ub complexes are stable in the presence of catalytic Cys mutant versions of HECT domains. Another mechanistic difference is reflected by exposure of Ub's Ile44 in HECT E3 intermediates, whereas for RING and related E3s the enzyme bearing the thioester-bound Ub binds the notorious 'I44 patch' on Ub to orient the thioester bond. Thus, different E3 families have evolved distinct approaches to activate Ub transfer to appropriate targets.

A major question is how lysines are selected for Ub and Ub-like protein modification. To date, this is best understood for E3-independent interactions between E2s and substrates, including Ub during polyubiquitination. These include Ubc9 binding a specific SUMO target sequence (*Bernier-Villamor et al., 2002*), Ube2S binding an acidic portion of Ub that completes its catalytic site (*Wickliffe et al., 2011*), and Ubc13 binding a partner protein that presents a specific Ub target lysine (*Eddins et al., 2006*). Mechanisms underlying RING E3 presentation of substrates are generally less understood. For example, different members of the SCF family of RING E3s appear to utilize distinct selections: context is a major determinant of target lysines during SCF$^{Cdc4}$-dependent ubiquitination of Sic1 in vivo, (*Sadowski et al., 2010*), spatial proximity to the noncovalent binding site is important for in vitro ubiquitination of β-catenin by un-neddylated SCF$^{β-TRCP}$ (*Wu et al., 2003*), and future studies will be required to determine mechanisms dictating substrate lysine selection by NEDD8-activated SCF E3s. Our data show how a NEDD4-family HECT E3's architecture prioritizes lysines recruited to a WW domain by their placement relative to a composite catalytic center that involves residues from both HECT domain lobes. Although future studies will be required to reveal how substrates recruited to other domains of HECT E3s are selected for ubiquitination, conservation of key residues indicates that the ligation mechanism used by Rsp5 broadly applies to a range of HECT E3s controlling numerous physiological processes (*Figure 9*).

Following Ub ligation, the HECT domain must reload with Ub, requiring C-lobe rotation to receive another Ub from E2. This may be stimulated by discharge of Ub from the HECT domain onto the

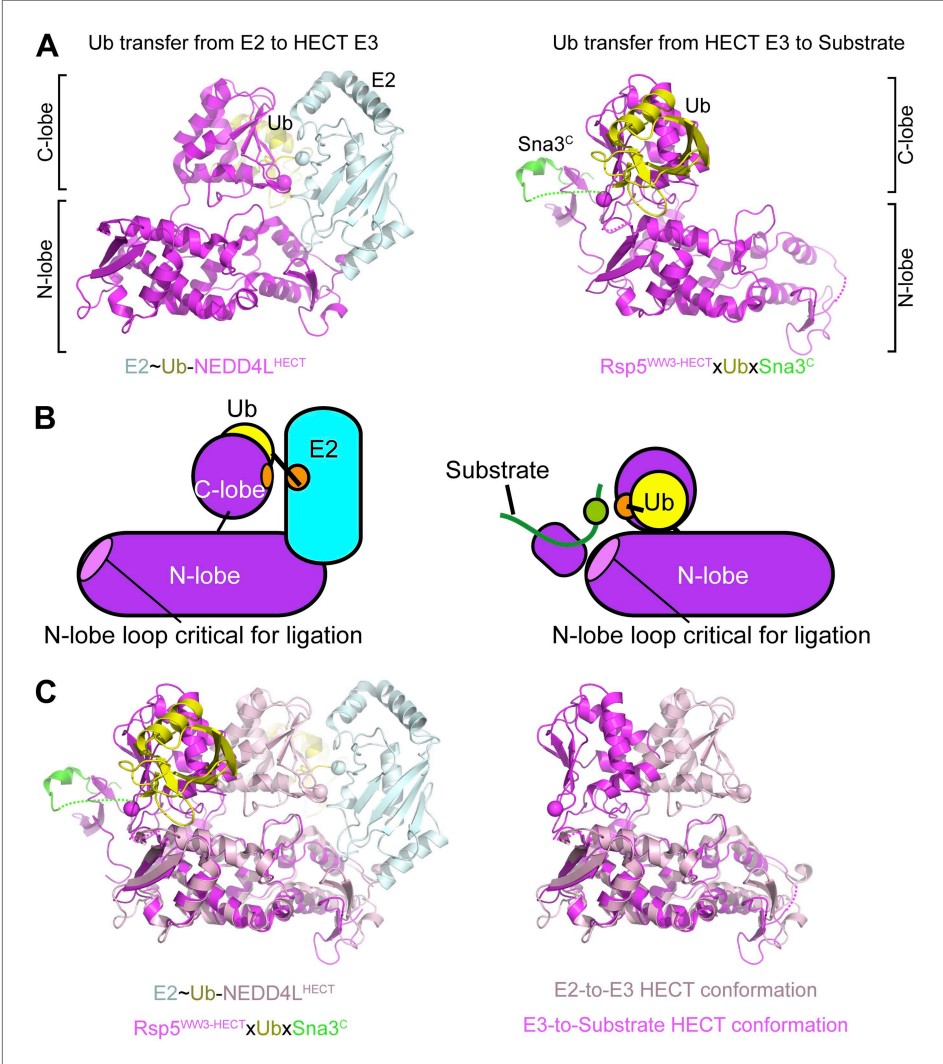

**Figure 11**. Structural view of the ubiquitin transfer cascade for a HECT E3. (**A**) Top left: Prior structure of E2 (pale cyan, with active site as sphere)~Ub (yellow)-NEDD4L[HECT] (violet, with active site as sphere) (***Kamadurai et al., 2009***). Right: new structure of Rsp5[WW3-HECT] (violet, with active site as sphere) xUb (yellow) xSna3[C] (green) aligned over the N-lobes of the NEDD4L and Rsp5 HECT domains. (**B**) Schematic views of E2-to-E3 Ub transfer and E3-to-substrate Ub ligation. (**C**) The different relative orientations of the HECT domain N-lobe with respect to the HECT domain C-lobe~Ub are highlighted by superimposing the N-lobes of the E2 (pale cyan, with active site as sphere)~Ub (yellow)-NEDD4L[HECT] (pink, with active site as sphere) (***Kamadurai et al., 2009***) and Rsp5[WW3-HECT] (violet, with active site as sphere) xUb (yellow) xSna3[C] (green) structures. The HECT domain portions of the two structures are shown on the right.

substrate, which would diminish the number of contacts at the junction between the HECT domain N- and C-lobes, allowing the C-lobe to both face E2 and separate from Ub. Notably, our mutational data also suggest utilization of a common HECT domain catalytic architecture for both substrate ligation and Ub chain formation (***Figures 2A and 8***). Although additional investigation will be required to reveal how Ub itself is positioned as an acceptor for polyubiquitination, we note that the HECT C-lobe region we found to interact noncovalently with the donor Ub determines ubiquitin chain lysine specificity in vitro (***Kim and Huibregtse, 2009***). We speculate that during polyubiquitination, the sum of weak interactions between HECT E3 domains and various substrates, adaptors, and regulators, sequentially drives the conformational pathway toward the active form at each step in HECT E3 cascades.

# Materials and methods

## Protein purification

Proteins and peptides used in this work are listed in *Table 1*. GST-tagged Ub and UbcH5B were expressed and purified as described (*Kamadurai et al., 2009*). His-tagged human E1 (*Berndsen and Wolberger, 2011*) and Ub were purified by nickel-affinity chromatography and gel filtration. Rsp5$^{WW3-HECT}$ used in structural studies was nickel-affinity-purified, treated with TEV, and purified by ion exchange. NEDD4, NEDD4L, Rsp5$^{WW3-HECT}$, and Rsp5$^{HECT}$ proteins used in the assays were pulled down by affinity chromatography and cleaved from their tags on-column with TEV. All other Rsp5 fragments were further purified by ion-exchange and gel filtration. All Sna3$^C$ proteins were HisMBP fusions, purified by nickel affinity. For assays, substrates either retained HisMBP or were separated from it by TEV treatment and gel filtration.

Proteins for crosslinking were treated with 10 mM DTT for 30 min and desalted into 25 mM HEPES 7.0, 150 mM NaCl. Rsp5$^{WW3-HECT}$xUbxSna3$^C$ was prepared by reacting single-Cys Rsp5$^{WW3-HECT}$ and Ub G75C with the Sna3$^C$-maleimide peptide in a molar ratio of 1:2:3, respectively, for 3 min on ice. The crosslinking reaction was terminated with excess DTT and the products were purified using ion exchange and sizing chromatography.

## Generation of Sna3$^C$ peptide with a homobifunctional sulfhydryl crosslinker

### Reagents

All reagents used were peptide synthesis grade. All Fmoc amino acids and N,N,N′,N′-tetramethyl-uronium-hexafluoro-phosphate (HBTU) were obtained from AnaSpec (San Jose, CA). Rink amide

**Table 1.** Constructs and peptides

| Constructs | Description |
|---|---|
| Rsp5 | |
| HisMBP-TEV-Rsp5$^{FL}$ | Rsp5 residues 1–809 were cloned into pRSF-1b with a His$_6$MBP tag, a poly Asn linker, a TEV site, and a GSGGS linker on the N-terminus |
| GST-TEV-Rsp5$^{WW1-3-HECT}$ | Rsp5 residues 217–809 cloned into pGEX-4T1 modified to include a TEV site and a GSGGS linker between GST and Rsp5 |
| GST-TEV- Rsp5$^{WW2-3-HECT}$ | Rsp5 residues 331–809 cloned as above |
| GST-TEV-Rsp5$^{WW3-HECT}$ | Rsp5 residues 383–809 cloned as above |
| GST-TEV-Rsp5$^{HECT}$ | Rsp5 residues 432–809 cloned as above |
| HisMBP-TEV-Rsp5$^{WW3-HECT}$ single-Cys | Rsp5$^{WW3-HECT}$ with the His$_6$MBP-polyN-TEV-GSGGS tag in pRSF-1b and C455L, C517G, and C721A mutations; retains C777; used in structure studies |
| NEDD4$^{WW1-4-HECT}$ | NEDD4 residues 165–900 cloned into pGEX-4T1 as above |
| NEDD4L$^{WW1,3-4-HECT}$ | NEDD4L residues 165–955 but lacking 356–399 (WW2) cloned as above |
| Sna3 | |
| HisMBP-TEV-Sna3$^C$ | Sna3 residues 71–133 with His$_6$MBP-polyN-TEV-GSGGS on the N-terminus |
| Sna3$^C$-maliemide peptide | Sna3 residues 104–128 chemically synthesized with azidohomoalanine at position 125, N-terminal acetylation, and C-terminal amidation |
| Biotin-Sna3$^C$ peptide | Same as above but with K125, N-terminal biotinylation, and C-terminal carboxylation |
| E2 | |
| UbcH5B | As described previously (*Kamadurai et al., 2009*) |
| E1 | |
| Human UBA1 | Human UBA1 with a non-cleavable His-tag on the N-terminus expressed in pET21d |
| Ub | |
| His-Ub-G75C | His-Ub terminating with G75C; used for structure |
| His-1Cys Ub | His-Ub with Cys N-terminal of Met1 for fluorescent labeling |
| GST-TM-2TK-Ub | Human Ub cloned into pGEX2TK (*Kamadurai et al., 2009*) |

MBHA resin and 6-chlorobezotriazole (6-chloro-HOBt) were purchased from AAPPTec, Inc (Louisville, KY). Piperidine, 4-methylmorpholine (NMM), diisopropylethylamine (DIPEA), acetic anhydride, phenol, triisopropylsilane (TIS), 2,5-dihydroxybenzoic acid (DHB), and thioanisole were obtained from Sigma-Aldrich (St. Louis, MO). Trifluoroacetic acid (TFA) was purchased from Life Technologies (Carlsbad, CA). Dimethylformamide (DMF), N-methylpyrrolidone (NMP), and acetonitrile were purchased from VWR International (Atlanta, GA). Diethyl ether was purchased from Fisher Scientific (Suwanee, GA). Fmoc-azidohomoalanine was purchased from Bachem (Torrance, CA).

## Synthesis of the Sna3$^C$ azidopeptide

Acetyl-AQPPAYDEDDEAGADVPLMDN QQ-NH$_2$
[M+H]
exact mass: 2656.1222

The azidopeptide was synthesized on a fully automated Liberty microwave peptide synthesizer (CEM Corp., Matthews, NC) at 100 µmol scale using Fmoc-Rink-Amide-Wang resin (loading 0.51 mmol/g). Deprotection was performed in two stages using 20% piperidine in NMP containing 0.1 M of 6-chloro-HOBt. Stage one of deprotection was for 30 s at 75°C followed by stage two which was for 3 min at 75°C. Coupling was performed using 0.2 mmol amino acids (5 eq., in DMF), 0.3 mmol HBTU (5 eq.), 0.3 mmol 6-chloro-HOBt (5 eq.), and 0.6 mmol NMM (10 eq.) in NMP for 10 min at 50°C. All amino acids were double coupled. A capping cycle using 4.75% acetic anhydride and 2.25% DIPEA in NMP was run after the introduction of each amino acid. Fmoc-azidohomoalanine was coupled similarly using an external amino acid bottle. After final deprotection, a capping cycle was run to introduce the N-terminal acetyl group. Cleavage was performed using TFA/phenol/TIS/EDT/thioanisole/water 82.5/2.5/2.5/2.5/5.0/5.0 for 2 hr at room temperature. The cleavage mixture was filtered and precipitated using ice cold diethyl ether. After centrifugation, the crude peptide was dissolved in water and lyophilized.

## Sna3$^C$ azidopeptide analysis/quality control

The crude peptide was dissolved in water and analyzed on a Waters 2695 analytical HPLC system (Waters, Milford, MA) using a C18, 5 µm, 250 × 4.6 mm column (Grace Vydac, Deerfield, IL), over 45 min with a flow rate of 1 ml/min, and using a gradient of 0–100% B, where buffer A is 0.1% TFA in water, and buffer B is water/acetonitrile/TFA 80/20/0.1. Detection was at 220 nm. Mass analysis was performed on a Bruker UltraFlex III TOF/TOF mass spectrometer (Bruker Daltonics, Billerica, MA) using DHB as the matrix.

## Sna3$^C$ azidopeptide purification

The crude peptide was dissolved in water and purified on a Waters 2695 semi-preparative HPLC system (Waters) using an XBridge C18, 5 µm, 250 × 10 mm column (Waters) over 45 min with a flow rate of 4 ml/min, and using a gradient of 0–40% B, where buffer A is 0.1% TFA in water, and buffer B is 0.1% TFA in acetonitrile. Detection was at 220 nm and 240 nm. Fractions containing the correct mass were analyzed on the analytical HPLC system, pooled, and lyophilized. The HPLC purity of the purified azidopeptide was >99%. The correct mass for the azidopeptide was observed at 2656.14 Da for M+H (calculated mass: 2655.15 Da).

## General chemical methods associated with attaching the crosslinker

All commercial reagents were used without further purification and the solvents were dried using the dry solvent system (Glass Contour Solvent Systems; SG Water, Nashua, NH). All reactions were monitored by thin-layer chromatography (TLC) carried out on EMD Chemicals silica gel 60-F 254 coated glass plates and visualized using I$_2$ or UV light (254 nm). Analysis by LC-MS was performed by using an XBridge C18 column run at 1 ml/min, and using gradient mixtures of (A) water (0.05% TFA) and (B) methanol. Low-resolution mass spectra (ESI) were collected on a Waters Micromass ZQ in positive-ion mode. Flash-chromatography was performed on a Biotage SP4 chromatography system using Biotage Flash KP-Sil and KP-NH amine pre-packed columns. Nuclear magnetic resonance (NMR) spectra were obtained on Bruker Avance II NMR spectrometer at 400 MHz for $^1$H-NMR spectra. Chemical shifts (ppm) are reported relative to TMS or the solvent peak. Signals are designated as follows: s, singlet; d, doublet; dd, doublet of doublet; t, triplet; q, quadruplet; m, multiplet. Coupling constants (*J*) are shown in Hertz.

## Preparation of 3,3′-(prop-2-yn-1-ylazanediyl)bis(propan-1-ol)

A solution of prop-2-yn-1-amine (0.349 ml, 5.45 mmol) and potassium carbonate (2.26 g, 16.3 mmol) in MeCN (15 ml) was stirred for 1 hr. 3-Bromopropan-1-ol (2.96 ml, 32.7 mmol) was added to the reaction mixture, heated to reflux, and stirred for 3 hr. Water (50 ml) was added to the reaction mixture and extracted with ethyl acetate (3 × 50 ml). The combined organic layers were washed with saturated brine, dried over $Na_2SO_4$, filtered, and concentrated. The crude product was purified by flash column chromatography (Biotage SP4, 40+M column, eluting with DCM/MeOH, 0–12% gradient) to give the desired product (303 mg, 33% yield): [1]H NMR (400 MHz, $CDCl_3$) δ 3.75 (t, $J$ = 5.7 Hz, 4H), 3.48 (d, $J$ = 2.4 Hz, 2H), 2.74 (t, $J$ = 6.4 Hz, 4H), 2.22 (t, $J$ = 2.4 Hz, 1H), 1.77–1.68 (m, 4H).

## Preparation of N,N-bis(3-azidopropyl)prop-2-yn-1-amine

DBU (1.36 ml, 9.01 mmol) and diphenylphosphorylazide (1.94 ml, 9.01 mmol) were added to a stirring solution of 3,3′-(prop-2-yn-1-ylazanediyl)bis(propan-1-ol) (257 mg, 1.501 mmol) in DMF (6 ml). The solution was heated to 100°C and stirred for 2 hr. The dark brown reaction mixture was added to water (20 ml) and extracted with ethyl acetate (3 × 20 ml). The combined organic layers were washed with saturated brine, dried over $Na_2SO_4$, filtered, and concentrated. The crude product was purified by flash column chromatography (Biotage SP4, 40+M column, eluting with DCM/MeOH, 0–10% gradient) to give a colorless oil (256 mg, 77% yield): [1]H NMR (400 MHz, $CDCl_3$) δ 3.38 (d, $J$ = 2.3 Hz, 2H), 3.35 (t, $J$ = 6.7 Hz, 4H), 2.57 (t, $J$ = 6.8 Hz, 4H), 2.19 (td, $J$ = 2.3, 1.1 Hz, 1H), 1.73 (p, $J$ = 6.8 Hz, 4H).

## Preparation of N1-(3-aminopropyl)-N1-(prop-2-yn-1-yl)propane-1,3-diamine

A solution of N,N-bis(3-azidopropyl)prop-2-yn-1-amine (256 mg, 1.16 mmol) and $PPh_3$ (668 mg, 2.55 mmol) in THF/$H_2O$ (9:1, 6.7 ml) was stirred overnight. The reaction mixture was concentrated and purified by flash column chromatography (Biotage SP4, 25+S KP-NH amine column, eluting with DCM/MeOH, 0–15% gradient) to give a yellow oil (132 mg, 67% yield): [1]H NMR (400 MHz, MeOD) δ 3.43 (d, $J$ = 2.2 Hz, 2H), 2.69 (t, $J$ = 7.1 Hz, 4H), 2.61–2.51 (m, 5H), 1.71–1.56 (m, 4H).

## Preparation of 1,1′-([prop-2-yn-1-ylazanediyl]bis[propane-3,1-diyl])bis(1H-pyrrole-2,5-dione)

N-Methoxycarbonylmaleimide (155 mg, 1.00 mmol) was added portionwise over the course of 15 min to a vigorously stirring solution of N1-(3-aminopropyl)-N1-(prop-2-yn-1-yl)propane-1,3-diamine (77 mg, 0.46 mmol) in saturated aqueous $NaHCO_3$ (2.5 ml) at 0°C. The ice bath was removed and the heterogeneous solution was stirred for 1 hr at room temperature. The reaction mixture was diluted with water (3 ml) and extracted with ethyl acetate (3 × 5 ml). The combined organic layers were washed with saturated brine, dried over $Na_2SO_4$, filtered, and concentrated. The crude product was purified by flash column chromatography (Biotage SP4, 12+S column, eluting with hexanes/ethyl acetate,

0–100% gradient) to give a white solid (90 mg, 60% yield): $^1$H NMR (400 MHz, CDCl$_3$) δ 6.69 (s, 4H), 3.57 (t, J = 7.2 Hz, 4H), 3.38 (s, 2H), 2.47 (t, J = 6.8 Hz, 4H), 2.13 (d, J = 2.3 Hz, 1H), 1.76–1.64 (m, 4H).

## Procedure for click chemistry (schematic shown below)

(+)-Sodium L-ascorbate (120 µmol) was added portionwise to a solution of 1,1′-([prop-2-yn-1-ylazanediyl]bis[propane-3,1-diyl])bis(1H-pyrrole-2,5-dione) (120 µmol), the azidopeptide (30 µmol), and CuSO$_4$ (120 µmol), in H$_2$O/tBuOH (2:1, 20 ml). The solution was stirred for 1 hr and then concentrated.

## Purification of the Sna3$^C$ peptide with a homobifunctional sulfhydryl crosslinker

The crude peptide was dissolved in water:acetonitrile (90:10) and purified on a Waters 2695 semi-preparative HPLC system using an XBridge C18, 5 µm, 250 × 10 mm column (Waters) over 45 min with a flow rate of 4 ml/min, and using a gradient of 0–50% B, where buffer A is 0.1% TFA in water, and buffer B is 0.1% TFA in acetonitrile. Detection was at 220 nm and 240 nm. Fractions containing the correct mass were analyzed on the analytical HPLC system, pooled, and lyophilized. HPLC purity of the purified peptide was >99%. The correct mass for the bis-maleimide peptide (peptide for crosslinking) was observed at 2985.42 Da for M+H (calculated mass: 2984.25 Da).

## Crystallization and structure determination

Rsp5$^{WW3-HECT}$x$^{MBP}$Sna3$^C$ (40.8 mg/ml) crystals were grown by hanging drop vapor diffusion in 0.1 M Bis-Tris propane pH 7.2, 21% wt/vol PEG3350, and 1% wt/vol polypropylene glycol at room temperature, and cryoprotected in 20% glycerol in the mother liquor. Diffraction data were processed using HKL2000 (*Otwinowski and Minor, 1997*). PHASER (*McCoy et al., 2007*) was used to implement molecular replacement, searching separately for Rsp5 N-lobe, Rsp5 C-lobe, and Ub for the two copies per asymmetric unit. The WW3 domain and Sna3 peptide were built manually and the model was subjected to multiple rounds of refinement and rebuilding using REFMAC (*Murshudov et al., 1997*) and Coot (*Table 2*; *Emsley et al., 2010*). The crosslinker and the Sna3 portion near it were not visible in the map. The final model contains Rsp5 (383–806, chain A; 384–806, chain B), Sna3 (105–112, chain C; 104–117, chain D), and Ub (1–75, chains E and F).

## Biochemical assays

Error bars are from triplicates of each experiment. Assays were performed with either radiolabeled (*Kamadurai et al., 2009*) or fluorescently labeled Ub and stopped by adding SDS loading buffer, unless specified, and products separated by nonreducing SDS-PAGE and visualized by fluorescent imaging with Typhoon 9200 or autoradiography. DTT-treated controls of either all samples or the final time-point in a time-course assay confirmed thioester-linked intermediates. To produce fluorescently labeled ubiquitin, 10× fluorescein-5-maleimide (Anaspec) was reacted with 1Cys Ub for 3 hr at room temperature. Excess label was quenched with DTT and removed through extensive desalting steps involving dialysis and PD-10 columns.

Pulse-chase assays to monitor Ub transfer from E2 to Rsp5 to Sna3$^C$ or Ub were performed in two steps. First, UbcH5B (9.1 µM) was reacted with E1 (0.42 µM) and fluorescent Ub (16.7 µM) in 50 mM Tris 7.6, 300 mM NaCl, 2 mM ATP, 10 mM MgCl$_2$, and 0.04 mg/ml of ovalbumin for 30 min at room temperature or 1 hr at 18°C for $^{32}$P-labeled Ub. These first reactions were quenched by diluting fourfold with 25 mM MES 6.5, 100 mM NaCl, 10 mM EDTA and desalting in the same buffer using Zeba Spin Desalting Columns (Pierce), or diluting fourfold with 25 mM HEPES 7.5, 100 mM NaCl, 25 mM EDTA. Second,

**Table 2.** Data collection and refinement statistics

| | Rsp5$^{WW3-HECT}$xUbxSna3$^C$ |
|---|---|
| **Data collection*** | |
| Space group | $P2_1$ |
| Cell dimensions | |
| a, b, c (Å) | 83.046, 78.923, 96.722 |
| α, β, γ (°) | 90, 101.67, 90 |
| Resolution (Å) | 30–3.1 (3.21–3.1)$^\dagger$ |
| $R_{sym}$ or $R_{merge}$ | 7.2 (36.1) |
| I/σI | 15.5 (2.0) |
| Completeness (%) | 94.9 (92.7) |
| Redundancy | 3.4 (3.2) |
| **Refinement** | |
| Resolution (Å) | |
| No. reflections | 21,323 |
| $R_{work}$/$R_{free}$ | 25.1/29.9 |
| No. atoms | |
| Protein | 7944 |
| Ligand/ion | 0 |
| Water | 0 |
| B-factors | |
| N-lobe (chains A, B) | 83.1, 88.2 |
| C-lobe (chains A, B) | 91.9, 109.7 |
| WW3 (chains A, B) | 100.0, 100.3 |
| Sna3$^C$ (chains C, D) | 123.5, 139.3 |
| Ub (chains E, F) | 124.7, 122.2 |
| Ligands | NA |
| Water | NA |
| R.m.s. deviations | |
| Bond lengths (Å) | 0.009 |
| Bond angles (°) | 1.05 |

*Data were collected from single crystals.

$^\dagger$Values in parentheses are for highest-resolution shell.

E2~Ub was mixed with excess Rsp5 and Sna3$^C$ on ice to initiate a single round of substrate ubiquitination. For Rsp5$^{WW3-HECT}$ mutants and different fragments of Rsp5, the final concentrations were ~0.4 µM E2~Ub, 2–2.7 µM E3, and 10–10.2 µM Biotin-Sna3$^C$ or 200 µM unlabeled Ub. Reactions were quenched at the indicated times and analyzed by fluorescent imaging or autoradiography (Storm phosphorimager, GE) of nonreducing SDS-PAGE gels. Thioester-linked intermediates were confirmed by DTT-reduced controls collected by treating half of the last time-point samples with 100 mM DTT. E3~Ub conjugates that accumulate for the defective mutants are thoroughly reduced with DTT except for the Lys432Ala/Arg433Ala/Asp434Ala/Arg436Ala and Arg437Ala/Lys438Ala/Ile440Ala/Tyr441Ala mutants (data not shown). Total activity within each lane was determined by adding all three intensities and was used to estimate the relative yield of each species (E2~Ub, Rsp5$^{WW3-HECT}$~Ub, and Sna3$^C$~Ub or Ub~Ub) and scaled with respect to wild-type proteins.

For substrate selection assays summarized in *Figure 10C* and *Figure 10—figure supplement 1*, 3.6 µM of Rsp5$^{WW1-3-HECT}$ and 6.4 µM of HisMBP-TEV-Sna3$^C$ were used. In this case, the reaction was terminated at 15 s with 100 mM DTT and products were incubated with TEV for 30 min at room temperature to liberate HisMBP. Following quenching and separating on SDS gels, each product (Sna3$^C$~Ub, Rsp5~Ub, and HisMBP~Ub) was quantified using ImageQuant TL. Total activity within each lane was determined by adding all three intensities and was used to estimate the relative yield of each species. These yields were then scaled in proportion to the total activity observed with wild-type proteins to obtain normalized product formation.

Rapid-quench flow kinetic studies were performed at 25°C by reacting 2 µM of E2~Ub with a mixture of 4 µM E3 and 30 µM Biotin-Sna3$^C$, using the KinTeK RQF-3 instrument in 25 mM MES 6.5, 100 mM NaCl, 10 mM EDTA. The reactions were quenched with SDS gel-loading buffer. E2~Ub was generated as described above but with 0.5 µM E1 and 10 µM each of UbcH5B and fluorescent Ub. Product separation by SDS-PAGE, visualization by fluorescent scanning, quantification, and normalization within each lane were performed as described for the substrate selection experiments. The yields were normalized across gels. Error bars are from triplicate measurements. Rates were obtained by fitting the product vs time data to linear functions using DeltaGraph (Red Rock Software).

All multiple turnover assays were performed at room temperature in 25 mM HEPES 7.5, 100 mM NaCl, 5–6 mM MgCl$_2$, 1 mM ATP, 0.2 mg/ml ovalbumin, 100 nM E1, 550 nM E2. The assay described in *Figure 1—figure supplement 1B* was performed with Rsp5$^{FL}$ (78 nM) and Rsp5$^{WW1-3-HECT}$ wild-type and WW mutants (70 nM), 5 µM wild-type or P107A MBP-free Sna3$^C$, and 5 µM fluorescent Ub for 2 min. The substrate ubiquitination assay in *Figure 6—figure supplement 1* was performed with 2.5 µM methylated fluorescent Ub, 50 µM Biotin-Sna3$^C$, and 1 µM of the indicated Rsp5$^{WW3-HECT}$ mutants. Autoubiquitination of NEDD4, NEDD4L, and Rsp5 (*Figure 9*) was assayed with 1 µM E3 and 2.5 µM fluorescent Ub for 2 min.

## Genetic assays for Rsp5-dependent yeast viability

Null *rsp5Δ* mutant (*MATα: leu2-3*, 112 *ura3-52*, *his3-Δ200*, *trp1-Δ901 lys2-801 suc2-Δ9 mel rsp5Δ::HIS3* + pNTAP416-Rsp5-WT) and *rsp5-1* mutant (LHY23: *MATa*: *rsp5-1 leu2-3, 112 ura3-52 bar1*) strains were described previously (*Dunn and Hicke, 2001*; *Oestreich et al., 2007*). A previously described centromere-containing low copy plasmid expressing N-terminally HA tagged wild-type *RSP5* (*Gajewska et al., 2001*) was converted to a *LEU2* plasmid (pPL4333) and used to create the mutants.

For experiments with *rsp5Δ* null mutants, single colony transformants were grown in minimal media lacking uracil and leucine to select for plasmids expressing both wild-type and mutant versions of *RSP5*. Cells (1 OD) grown to $OD_{600} = 1.0$ were resuspended in water, serially diluted 1:10, and spotted onto SC-Leu plates with or without 1 mg/ml 5-fluoroorotic acid (5-FOA) and grown at 30°C for 2–5 days. For experiments with *rsp5-1* mutants, single colony transformants were similarly serially diluted and spotted onto YPD and grown for 2–5 days at 30°C or 37°C. To immunoblot for HA-Rsp5, cells were grown to $OD_{600} = 1.0$, centrifuged, incubated with 0.2 M NaOH for 3 min, and resuspended in Laemmli sample buffer containing 8 M urea. Samples were analyzed by SDS-PAGE followed by immunoblotting using anti-HA and anti-PGK antibodies.

## Structural modeling of the Rsp5[WW3-HECT]xUbxSna3[C] catalytic center

We used various modules in the Rosetta3 modeling suite to construct the different features of the catalytic center; modified versions of the UBQ_E2_thioester and FloppyTail algorithms will be available in the next Rosetta3 release (currently 3.4) (*Leaver-Fay et al., 2011*). The UBQ_E2_thioester application generates a thioester-bonded complex and samples torsion angles for the thioester and nearby residues, including cysteine chi1 and 2, Gly76 phi and pseudo-psi, Gly75 phi and psi, Arg74 phi and psi, and Leu73 phi and psi (*Saha et al., 2011*). Additionally, this code was modified to model the location of terminal Ub acceptor, positioning the NZ of the acceptor lysine collinear with the carbonyl bond and 2.4 Å from the carbonyl carbon on the C-terminal glycine of Ub. Input structures for Rsp5, Ub, and Sna3 were obtained from the Rsp5[WW3-HECT]xUbxSna3[C] co-crystals. Several constraints were placed on the Ub position derived from the Rsp5[WW3-HECT]xUbxSna3[C] three-dimensional structure. We focused on recovering the β-strand pairing between the Ub tail and the C-lobe active site. To precisely position the Ub tail, we introduced two angle constraints between the backbones of Ser774 of Rsp5 and Leu74 of Ub. As a preliminary step, we modeled five-residue Ub tails to determine the range of thioester torsion angles that could accommodate the β-sheet pairing and applied these angles to the starting pose. Additionally, we decreased the size of the moves on the cysteine chi1 and thioester bond to obtain more models with optimal geometry; even with this change models with various cysteine chi1 and thioester torsion angles were recovered. We filtered the subsequent models on constraint energy, RMSD to Ub, and visual inspection, and selected the top five models.

For these five models, we used the FloppyTail application to sample possible conformations of the Sna3 substrate (*Kleiger et al., 2009*). Constraints were placed on the location of the PPXY motif. Residues 115–124, between the PPXY motif and the acceptor lysine, were allowed to sample backbone torsion angles, and the models were filtered by lowest energy and RMSD to the crystal PPXY motif.

To ascertain possible N-lobe loop conformations for residues 491–495, we grafted a homologous segment of NEDD4L (PDB 3JW0) containing residues 630–645, which correspond to residues 483–505 on Rsp5. We used a two-step protocol to close and refine the loop. First, we used perturb KIC to close the loop in the absence of the Sna3 peptide and substituted the corrected amino acid identities that varied in the grafted NEDD4L loop. We allowed backbone sampling for residues 488–498, and selected the lowest energy closed loop; about 5% of the models had successfully closed N-lobe loops. Next, we refined the loop in the presence of the Sna3[C] peptide, both with and without a 3 Å distance constraint between the acceptor lysine and Asp495 side chain. Although the loop could adopt multiple orientations that could bring these residues in close proximity, without this constraint Rosetta favored orientations that placed Asp495 more distant from the acceptor lysine. We examined the models scoring in the 90th percentile and found that about half of them had a very similar loop orientation and choose a representative structure from this set.

## Acknowledgements

We are grateful to DW Miller, S Bozeman, J Rogers, and J Bollinger for administrative/computational support, to C Wolberger for the human UBA1 clone, D King for mass spectrometry support, and to members of the Schulman Lab, K Guy, and JW Harper for helpful discussions.

# Additional information

## Funding

| Funder | Grant reference number | Author |
|---|---|---|
| Howard Hughes Medical Institute | 045102 | Brenda A Schulman |
| National Institutes of Health | R01GM069530 | Brenda A Schulman |
| National Institutes of Health | 5P30CA021765 | Brenda A Schulman |
| ALSAC | | Brenda A Schulman |
| National Institutes of Health | R01GM073960 | Brian Kuhlman |
| National Institutes of Health | 5P41RR015301-10 | Igor Kurinov |
| National Institutes of Health | 8P41 GM103403-10 | Igor Kurinov |
| DOE | DE-AC02-06CH11357 | Igor Kurinov |
| National Institutes of Health | 5R01 GM058202 | Robert Piper |
| American Heart Association | | Hari B Kamadurai |

The funders had no role in study design, data collection and interpretation, or the decision to submit the work for publication.

## Author contributions

HBK, Conceived and oversaw the study, generated reagents, performed crystallographic and biochemical experiments, and prepared the manuscript with contributions from all authors; YQ, Performed mutational analysis and biochemical experiments, and prepared the manuscript with contributions from all authors; AD, Generated, crystallized, and improved crystals for Rsp5^WW3-HECT^xUbxSna3^C^, and laid the groundwork for biochemical assays; JSH, Performed computational modeling and edited the manuscript; CM, Made constructs and strains, performed yeast genetic experiments, and edited the manuscript; MA, PR, Designed, synthesized, and purified the Sna3^C^ peptide with a homobifunctional sulfhydryl crosslinker; DJM, Refined crystal structures, and analyzed and interpreted the data; JS, Developed expression systems and assays for the Rsp5 HECT domain; SML, Made algorithmic changes to Rosetta required for modeling the Rsp5~Ub-Sna3C catalytic center; IK, Assisted with crystallography, analysis, and interpretation of data; NF, Oversaw generation of the peptide with a homobifunctional sulfhydryl crosslinker; MH, Performed biophysical analyses on Rsp5 complexes; RP, Oversaw yeast genetics experiments, and helped prepare the manuscript; BK, Oversaw computational modeling; BAS, Provided guidance and oversight, assisted with crystallographic experiments and with designing and interpreting biochemical data, and prepared the manuscript with contributions from all authors

# Additional files

## Major dataset

The following dataset was generated:

| Author(s) | Year | Dataset title | Dataset ID and/or URL | Database, license, and accessibility information |
|---|---|---|---|---|
| Kamadurai HB, Miller D, Schulman BA | 2013 | Structure of an Rsp5xUbxSna3 complex: Mechanism of ubiquitin ligation and lysine prioritization by a HECT E3 | 4LCD; http://www.rcsb.org/pdb/search/structidSearch.do?structureId=4LCD | Publicly available at the RCSB Protein Data Bank (http://www.rcsb.org/). |

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
