## [Decision Letter]

Thank you for sending your work entitled “Mechanism of ubiquitin ligation and lysine prioritization by a HECT E3” for consideration at *eLife*. Your article has been favorably peer reviewed by 3 reviewers, one of whom is a Senior editor.

The Senior editor and the other reviewers discussed their comments before we reached this decision, and the Senior editor has assembled the following comments to help you prepare a revised submission.

Overall summary:

This is a very interesting paper in which the authors dissect the mechanism of a HECT E3 ligase. Ubiquitylation is an essential posttranslational modification that controls many aspects of cellular behavior. HECT-type E3 ubiquitin ligases were the first E3s to be discovered, and they are known to play many important roles in cellular signaling. HECT-E3s often catalyze monoubiquitylation reactions, during which they specifically modify a lysine residue in a substrate with ubiquitin; alternatively, HECT-E3s are also able to decorate substrates with ubiquitin chains, requiring an initial modification of a substrate lysine and subsequent repetitive modification of specific ubiquitin lysine residues. How HECT-E3s catalyze transfer of ubiquitin to a lysine residue in the substrate has remained poorly understood.

Using the essential HECT-E3 Rsp5, Schulman and colleagues provide structural insight into this important reaction. By determining the Lys preference in Rps5-substrates, as well as by alanine scanning of most of the HECT-domain, they first provide convincing evidence that a specific orientation of N- and C-lobe, and the substrate-binding WW-domain of Rsp5 is important for ubiquitin transfer to the substrate. A key advance reported in the paper is the clever use of an artificial residue that was incorporated in the substrate peptide, replacing the lysine that is normally present. The artificial residue incorporated a dual-headed maleimide, which allowed simultaneous crosslinking of the Cys residue of the HECT E3 as well as the terminal residue of the ubiquitin. This allowed stabilization of the otherwise transient complex between the substrate and the charged E3, and crystallographic visualization of the complex.

Work from the Schulman lab had previously shown a HECT E3 to bind to E2∼Ub in a conformation that places the HECT active site cysteine far from the substrate binding site. The new structure reported here is dramatically different, and shows that the Rsp5 HECT E3 undergoes a dramatic conformational change that brings the C-lobe of the HECT E3 with bound ubiquitin into a position so that the HECT∼Ub thioester in the C-lobe is brought near to the bound Sna3 substrate.

The structural results provide nice confirmation of the conclusion reached by the authors using detailed biochemistry, which is that in contrast to the Cys-carrying C-lobe of the ligase being a completely flexible, it prefers a particular orientation with respect to lysine presented by the substrate. The model presented by the authors is that the HECT E2 switches between two states, one for charging by E2 and one for transferring the ubiquitin to the substrate. The structure determined by the authors shows that the docking of the substrate to a WW domain attached to the HECT domain places steric constraints on the distance of the ubiquitylated lysine from the WW docking site. This is consistent with their biochemical analysis, which shows that if the lysine is too close to the WW docking site then ubiquitylation is inefficient. The authors then validate their structure through multiple approaches, including the rescue of mutants in the C-lobe through mutation of opposing residues in the N-lobe, and using a point mutant equivalent to a key mutation in Angelman’s Syndrome, they underscore the medical relevance of their studies. The Lys residue modified in Sna3, as well as a loop in Rsp5 close to its active site were not resolved in their structure, so the authors used modeling approaches based on their structure to probe the orientation of these residues.

Major concerns:

1) There is a serious concern that this paper might be unnecessarily difficult to read for people who are not experts in the ubiquitin field. This has to be fixed before the paper can be accepted. One indication of the problem faced by a general reader is the very short Introduction (two paragraphs) that falls short of setting the stage for the general reader. In addition to introducing the field and the principal question, the Introduction should bring the reader up to speed with the basic chemistry of the ubiqutinylation reactions. As it stands, the general reader will be lost by comments such as “the HECT domain forms a DTT-reducible thioester-linked intermediate” and the arguments made by the paper will be missed. The paper presents an enormous amount of data and digesting this could also obscure some of the most important results. The authors have chosen to present the mutagenesis data first and the structural results second. While the final presentation order should be as the authors think best, if they stick with this order then a properly written Introduction could help guide the reader through the results that follow. As currently written, the paper requires the reader to go back-and-forth to understand the logic.

2) The chemical trick used by the authors to trap the 3-way complex is both an asset and a potential pitfall of the structure. While perhaps unlikely, given all the mutagenesis data presented here, it is a potential caveat of analyzing crosslinked structures that the crosslink stabilizes an usually underpopulated conformation. It is indeed very interesting that the C-lobe undergoes such a large conformational change between charging by the E2 and transfer to the substrate, which is all the more surprising given how highly transient the HECT∼Ub intermediate is in the absence of substrate. The authors have a significant burden of proof since this artificial linkage could be what determines the relative position of the N- and C-lobe. Given this, all of the key data supporting the structural model should be integrated into the main text. For example, Figure 3–figure supplement 2 maps some of the alanine scanning mutants to the N-lobe-C-lobe interface and also shows the R501A/E502A experiment, which similarly addresses the validity of this interface. This figure should be in the main part of the paper as it really contains very important support for the model. Do the authors have data concerning the efficiency of the crosslink between the substrate peptide and Rsp5, and whether it mimics the Lys preference results (i.e., low efficiency if crosslink is attached to a residue within 9 amino acids from the ppxy motif)?

3) Some of the mutant analysis was done with an assay using a large fusion protein consisting of MBP fused to the Sna3^C^ peptide by a TEV cleavage site. The ubiquitinated substrate is then cleaved with TEV, which makes it possible to distinguish between Sna3 ubiquitination and MBP ubiquitination. What is the evidence that the surface mapped by the authors as contributing to HECT to substrate transfer (Figure 1–figure supplement 3) disrupt interactions with the C-lobe (i.e., support a conformational change) rather than with MBP (which would then not be relevant to a conformational change). A few of the alanine scan mutants were also assayed for diubiquitin formation (Figure 2—figure supplement 1), which seemed to give somewhat different results than the MBP fusion and suggest that the results are reporting as much on the substrate as on the HECT conformational change. These issues need clarification.

4) Rosetta was used to model the native complex based on the 3-way cross-linker. The discussion of the linkage to the active site, including statements about the rotamer of the catalytic cysteine, etc., are very detailed and more appropriate for a high-resolution crystal structure rather than modeling based on a moderate-resolution structure containing a chemical cross-linker. This part of the paper is not that critical – a more qualitative model would have sufficed.

---

## [Author Response]

*1) There is a serious concern that this paper might be unnecessarily difficult to read for people who are not experts in the ubiquitin field. This has to be fixed before the paper can be accepted. One indication of the problem faced by a general reader is the very short Introduction (two paragraphs) that falls short of setting the stage for the general reader. In addition to introducing the field and the principal question, the Introduction should bring the reader up to speed with the basic chemistry of the ubiqutinylation reactions. As it stands, the general reader will be lost by comments such as “the HECT domain forms a DTT-reducible thioester-linked intermediate” and the arguments made by the paper will be missed. The paper presents an enormous amount of data and digesting this could also obscure some of the most important results. The authors have chosen to present the mutagenesis data first and the structural results second. While the final presentation order should be as the authors think best, if they stick with this order then a properly written Introduction could help guide the reader through the results that follow. As currently written, the paper requires the reader to go back-and-forth to understand the logic*.

To address the review comments and improve the presentation of our work, we substantially rewrote the manuscript and improved the figures. For example:

• Figure 1 now introduces the basic chemistry of the reactions for HECT E3s.

• We expanded the Introduction to six paragraphs, including (1) a general description of ubiquitin transfer cascades with a more broad discussion E3 enzymes and the chemical reaction they catalyze; (2) an introduction to the NEDD4 family of HECT E3s and Rsp5, in particular; and (3) the rationale behind choosing Rsp5 and Sna3 as a model system for studying HECT E3-substrate interactions.

• In describing the first ubiquitin ligation reactions, we more clearly explain the experiments and expectations. Examples of new sentences added for clarity are: “Thioester-linked intermediates were confirmed by their susceptibility to reduction by DTT (Figure 1—figure supplement 1). Complexes in which Ub was linked via an isopeptide bond via autoubiquitination of Rsp5 or to a lysine on Sna3^C^ were not susceptible to reduction.” We also added more detailed explanations of how Ala scan and deletion mutagenesis results are interpreted.

• For clarity, we also moved the linker deletion mutational data to the end of the manuscript.

*2) The chemical trick used by the authors to trap the 3-way complex is both an asset and a potential pitfall of the structure. While perhaps unlikely, given all the mutagenesis data presented here, it is a potential caveat of analyzing crosslinked structures that the crosslink stabilizes an usually underpopulated conformation. It is indeed very interesting that the C-lobe undergoes such a large conformational change between charging by the E2 and transfer to the substrate, which is all the more surprising given how highly transient the HECT∼Ub intermediate is in the absence of substrate. The authors have a significant burden of proof since this artificial linkage could be what determines the relative position of the N- and C-lobe. Given this, all of the key data supporting the structural model should be integrated into the main text. For example, Figure 3–figure supplement 2 maps some of the alanine scanning mutants to the N-lobe-C-lobe interface and also shows the R501A/E502A experiment, which similarly addresses the validity of this interface. This figure should be in the main part of the paper as it really contains very important support for the model. Do the authors have data concerning the efficiency of the crosslink between the substrate peptide and Rsp5, and whether it mimics the Lys preference results (i.e., low efficiency if crosslink is attached to a residue within 9 amino acids from the ppxy motif)*?

We substantially rearranged our description of the mutational data supporting the active conformation. In particular, we describe the effects of the R501A/E502A and R559A/R560A/F561A mutants from the Ala scan in the main text. Additional data validating this interface include in vitro and in vivo compensation of the F806L mutation in the C-lobe with a complementary L506F mutation in the N-lobe. We do not have data concerning crosslinking efficiency.

*3) Some of the mutant analysis was done with an assay using a large fusion protein consisting of MBP fused to the Sna3*^*C*^
*peptide by a TEV cleavage site. The ubiquitinated substrate is then cleaved with TEV, which makes it possible to distinguish between Sna3 ubiquitination and MBP ubiquitination. What is the evidence that the surface mapped by the authors as contributing to HECT to substrate transfer (Figure 1–figure supplement 3) disrupt interactions with the C-lobe (i.e., support a conformational change) rather than with MBP (which would then not be relevant to a conformational change). A few of the alanine scan mutants were also assayed for diubiquitin formation (*Figure 2—figure supplement 1*), which seemed to give somewhat different results than the MBP fusion and suggest that the results are reporting as much on the substrate as on the HECT conformational change. These issues need clarification*.

We are very sorry that our initial description of the assays was confusing. We made several changes for clarity. First, the Ala scan was done with a peptide substrate. MBP-fusions were not used in these assays. Thus, MBP did not impact the conclusions from these assays. We now show a schematic view of the assays with peptide substrates in Figure 1. Second, the MBP fusion is *only* used in the substrate selection assay. We recognize that describing this assay first, as in the original manuscript, was confusing. In the revision, the substrate selection is the last assay presented. We hope the new position for this assay will alleviate confusion. Third, for ease of comparison, we decided to complete testing the entire panel of Ala mutants in the di-Ub synthesis assay, and to include the data in the main text as Figure 8. This allows the reader to make a more direct comparison of the bar graphs in the different figures. The conclusions remain the same with the full suite of mutants.

*4) Rosetta was used to model the native complex based on the 3-way cross-linker. The discussion of the linkage to the active site, including statements about the rotamer of the catalytic cysteine, etc., are very detailed and more appropriate for a high-resolution crystal structure rather than modeling based on a moderate-resolution structure containing a chemical cross-linker. This part of the paper is not that critical – a more qualitative model would have sufficed*.

We appreciate the concerns of the reviewers. In response to the suggestions, we revised the section entitled “A composite HECT domain catalytic center involving both HECT C-lobe and N-lobe” to address the concerns and to improve readability. Nonetheless, we do believe it is appropriate to include atomic-level detail in the modeled figures. We took great care to reproduce an atomic-level accurate model of the thioester linked Ub tail and, to our knowledge, this is the most accurate snapshot of the E3 ligase poised to transfer Ub to a substrate lysine. Additionally, the strategy outlined in this manuscript is generalizable and robust. The accuracy of our prediction is underscored by the striking similarity to a disulfide-Ub linked NEDD4 that was recently reported while this paper was in revision (Maspero et al., Nat. Struct. Mol. Biol. 20: 696–701, 2013; Figure 18).Author response image 1.Overlay of modeled thioester conjugated tail with recently disulfide conjugated Ub to NEDD4 (4BBN).A Rosetta model is depicted in pink, the Rsp5xUbXSna3C structure is green, and the disulfide chemical surrogate is colored grey. The Rosetta model provides near atomic-level accuracy of the thioester conjugated Ub tail. The similarities between the structure and the model strongly support the position of the acceptor lysine.